# A zircon case for super-wet arc magmas

Chetan Nathwani [1,2] ✉, Jon Blundy [3], Simon J. E. Large[2],
Jamie J. Wilkinson [2,4], Yannick Buret [2], Matthew A. Loader[2],
Lorenzo Tavazzani[1] & Cyril Chelle-Michou [1]

Arc magmas have higher water contents (2-6 wt.% $H_2O$) than magmas generated in other tectonic environments, with a growing body of evidence suggesting that some deep arc magmas may be 'super-wet' (>6 wt.% $H_2O$). Here, we use thermodynamic modelling to show that the behaviour of zirconium during magmatic differentiation is strongly sensitive to melt water contents. We demonstrate that super-wet magmas crystallise zircon with low, homogeneous titanium concentrations (75[th] percentile <10 ppm) due to a decrease in zircon saturation temperatures with increasing melt $H_2O$. We find that zircon titanium concentrations record a transition to super-wet magmatism in Central Chile immediately before the formation of the world's largest porphyry copper deposit cluster at Río Blanco-Los Bronces. Broader analysis shows that low, homogeneous zircon titanium concentrations are present in many magmatic systems. Our study suggests that super-wet magmas are more common than previously envisaged and are fundamental to porphyry copper deposit mineralisation.

The dominant volatile in arc magmas is water (4 wt.% average $H_2O$ in basaltic magmas)[1]. As these hydrous magmas differentiate and ascend through the crust, they saturate in a fluid phase which can trigger explosive volcanism or generate economically valuable porphyry Cu deposits[2–4]. The high water content influences chemical differentiation of magmas, and may thus be responsible for the bulk andesitic composition of modern continental crust[4]. Measurements of water contents in glassy melt inclusions suggest that most erupted arc magmas, ranging in composition from basalt to rhyolite, contain 0–6 wt.% $H_2O$ (Fig. 1), consistent with storage under water-saturated conditions at upper crustal (<15 km) depths[5,6]. This restricted range of water contents in arc-related melt inclusions, regardless of the extent of chemical differentiation, is at odds with the water contents produced by fractional crystallisation of a typical primitive basalt composition (4 wt.% $H_2O$), which should yield much higher $H_2O$ contents of 6–11 wt.% $H_2O$ due to the incompatible behaviour of $H_2O$ during crystallisation[7–9] (Fig. 1).

Several studies have questioned the reliability of melt inclusions because they are trapped in crystals formed at shallow depths and thus record post- or syn-degassing water contents[10]. The fidelity of the melt inclusion volatile record is further complicated because melt inclusions with melt $H_2O$ > 6 wt.% are more difficult to quench[11], and there is also the possibility of $H_2O$ loss via H diffusion through their host minerals and formation of $CO_2$-rich shrinkage bubbles[12,13]. The phenocryst record may also be compromised by crystal resorption during ascent from depth, and/or re-equilibration at shallow depths (e.g., estimated amphibole crystallisation depths cluster at 200–300 MPa[14,15]); obscuring any potential deep record of arc magma evolution at higher water contents.

Alternative methods provide a growing body of evidence for the formation of 'super-wet' magmas (>6 wt.% $H_2O$) at depth in arcs, such as from the high electrical conductivity of mid-crustal magma reservoirs[16], high volatile contents in lower-crustal cumulates[17], suppression of plagioclase crystallisation in some magmas[18] and comparisons of cumulate mineral assemblages with experimental observations[19–21]. Although super-wet arc magmas may be abundant at depth, they are not expected to persist to shallow depths with their high original water contents intact. During fluid-saturated ascent they will devolatilise and crystallise rapidly, because of the effect of $H_2O$ on lowering liquidus temperatures and the increase in viscosity as $H_2O$

[1]Department of Earth and Planetary Sciences, ETH Zürich, Zürich, Switzerland. [2]London Centre for Ore Deposits and Exploration (LODE), Natural History Museum, South Kensington, London, UK. [3]Department of Earth Sciences, University of Oxford, Oxford, UK. [4]Department of Earth Science and Engineering, Imperial College London, South Kensington, London, UK. ✉e-mail: chetan.nathwani@eaps.ethz.ch

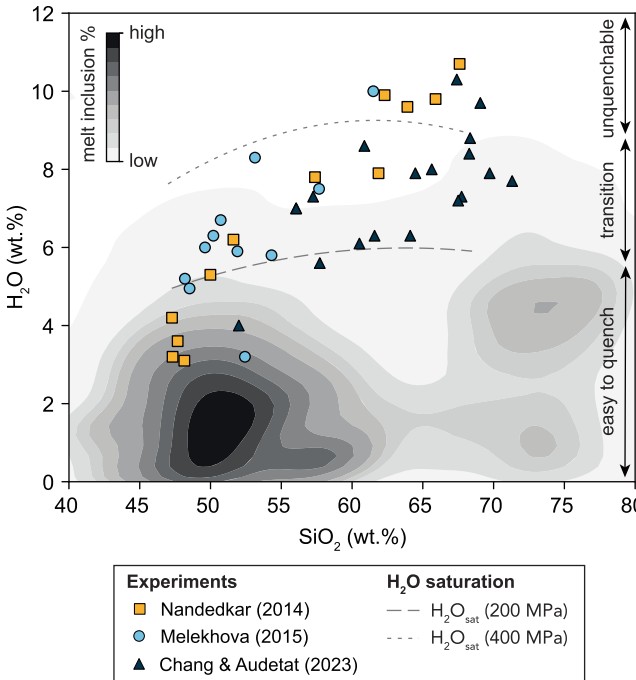

**Fig. 1 | Comparison of $H_2O$ contents of arc melt inclusions with glasses from high-pressure experiments.** Contour plot showing arc melt inclusion water contents as a function of silica content (derived from the GEOROC database). These water contents are markedly lower than those measured in high-pressure fractional/equilibrium crystallisation experiments using arc basalts (refs[7–9]; symbols). Dashed and dotted lines show the maximum water content dissolvable at 200 MPa and 400 MPa, respectively, assuming pure $H_2O$ liquid (calculated using MagmaSat on experimental liquids[7,39]). Vertical arrows indicate the quenchability ranges given by experiments whereby melt inclusions with 2–6 wt.% $H_2O$ are easy to quench to a glass, whereas those with higher $H_2O$ contents become increasingly difficult and eventually impossible to quench[11]. These data suggest that the melt inclusions database is biased towards low water contents due to both quenchability issues and shallow pressure crystallisation.

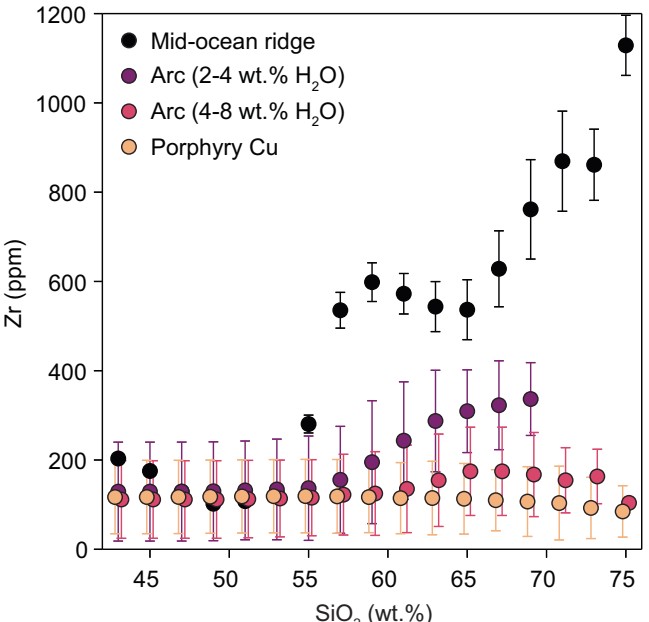

**Fig. 2 | Zirconium systematics for magmas with a range of $H_2O$ contents.** Whole-rock Zr systematics for mid-ocean ridge lavas[35], arc magmas and porphyry Cu deposit-related igneous suites. For arc magmas, the compositions are separated by lower (2–4 wt.%) and higher (4–8 wt.%) $H_2O$ where dissolved $H_2O$ is estimated based on the maximum of available melt inclusion data or from experimental constraints. Points show the binned mean of Zr concentrations per 2 wt.% $SiO_2$ and 2 s.e. uncertainties. No data could be found for arc rhyolites (>70 wt.% $SiO_2$) with 2–4 wt.% $H_2O$. Source data are provided as a Source Data file.

exsolves[22]. Consequently, 'super-wet' arc magmas would normally be expected to stall at deeper crustal levels[19]. However, several studies have inferred the presence of super-wet magmas in some volcanic centres and mid to upper crustal intrusions suggesting these magmas may occasionally ascend to the near surface[19,20,23–25].

Modelling studies have predicted that higher $H_2O$ contents in magmas enhance their ability to exsolve and outgas large quantities of metal-charged fluid[26–28]. Thus, the ascent of super-wet magmas could deliver large volumes of mineralising fluids to the shallow crust, which may be predisposed to form magmatic-hydrothermal ore systems[17,26,27,29] or be degassed to Earth's atmosphere[3,26]. Conversely, failure to ascend would increase the plutonic-volcanic ratio of arc magmatic systems and favour deeper release of volatiles, potentially hydrating the overlying crust[17,19] and leading to water-fluxed partial melting[30].

One mineral with potential to retain a record of deep magmatic conditions by surviving ascent from depth and escaping re-equilibration at shallow crustal levels is zircon. The conventional model of zircon saturation purports that the majority of zircon in calc-alkaline arc rocks are derived from shallow crystallisation (5–15 km) in intermediate to felsic magmas after significant crystallisation has elevated the concentration of Zr (an incompatible element prior to zircon saturation) sufficiently that zircon becomes saturated[31]. However, studies of exhumed crustal sections have revealed that significant zircon can be found in both deep and shallow portions of trans-crustal magmatic systems, with timings and durations of crystallisation at different depths that are indistinguishable when dated with high-

precision geochronology[32,33]. Some workers have also noted the presence of multiple zircon populations within single intrusions, inferred as crystal cargo from multiple storage depths[34]. Therefore, zircon sampled in upper crustal rocks may in some instances reflect polybaric arc magma differentiation and be a prime candidate to record early, super-wet stages of arc magma differentiation.

A global compilation of whole-rock compositions indicates that primitive (basaltic) magmas in both mid-ocean ridge (MOR) and arc environments have overlapping initial Zr concentrations (ca. 100 ppm Zr; Fig. 2). This suggests a limited role of source processes in controlling Zr systematics across these tectonic settings and that it is only through differentiation that these magmas depart from their common starting point (Fig. 2). In MOR magmas, Zr increases during differentiation by an order of magnitude due to crystal-melt segregation under zircon-undersaturated conditions[35]. However, in arc magmas, Zr increases more gradually with differentiation, showing an inflection late in magma differentiation due to zircon saturation[31].

The lower Zr concentrations of arc magmas at a given $SiO_2$ content have been attributed previously to their higher dissolved $H_2O$ concentrations relative to MOR magmas[31,35]. Considering arc lavas where $H_2O$ has been constrained, we find higher dissolved $H_2O$ (4–8 wt.%) causes the increase in Zr with differentiation to become even further suppressed compared to arc lavas with lower $H_2O$ (2–4 wt.%; Fig. 2). The extreme end-member of this series is represented by magmas from porphyry Cu systems, where Zr appears to be virtually invariant with magma differentiation, which has previously been suggested as evidence for differentiation under very wet conditions[36].

In this study, we test the ability of zircon to record super-wet conditions in arc magmas. We adopt a thermodynamic modelling approach coupled with zircon saturation models to show low zircon solubility under super-wet magmatic conditions leading to low, homogeneous Ti concentrations in zircon. We then use zircon Ti concentrations to track the emplacement of super-wet magmas in the

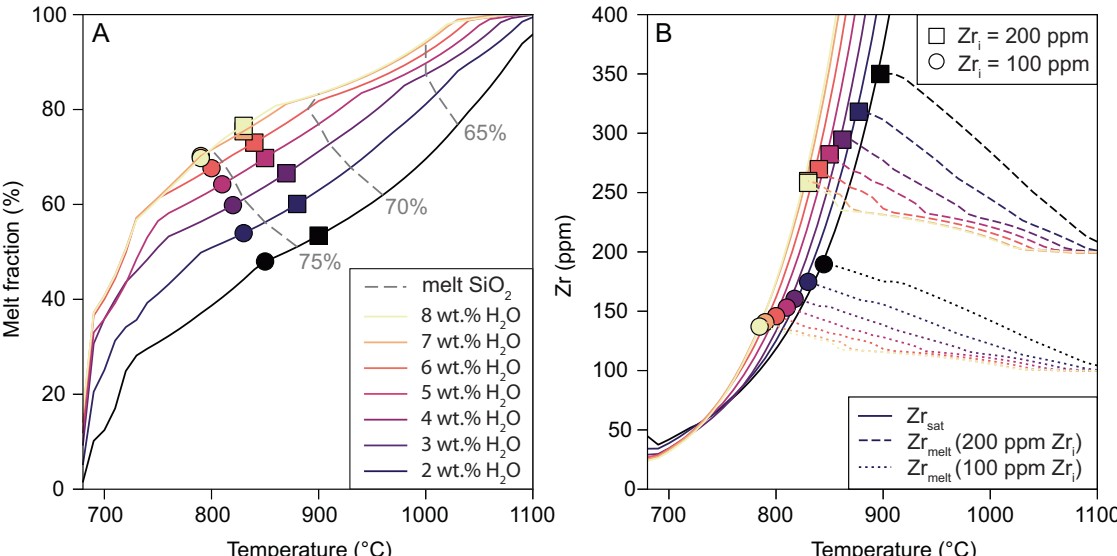

**Fig. 3 | Thermodynamic modelling of zircon saturation as function of initial melt $H_2O$ content.** Rhyolite-MELTS modelling of zircon saturation for an andesitic starting composition at 400 MPa for initial water contents between 2 and 8 wt.%. **A** Temperature vs melt fraction relationship; symbols indicate where zircon saturates given $Zr_i$ of 100 ppm (circles) and 200 ppm (squares). Grey dashed contours indicate when residual melts reach 65, 70 and 75 wt.% $SiO_2$ (calculated on an anhydrous basis). **B** The concentration of Zr in residual melt ($Zr_{liquid}$) during melt cooling and crystallisation with dashed and dotted lines showing the trajectories for 200 ppm and 100 ppm $Zr_i$, respectively. The mineral-melt partition coefficients for Zr ($D_{Zr}$) were compiled and parameterised as a function of temperature (see "Methods" section). Solid lines show the temperatures and Zr concentrations at which zircon saturates for each rhyolite-MELTS model with different water contents (A for legend). Zircon saturation occurs where dashed/dotted and solid lines of the same colour intersect (marked by square/circle symbols). Subsequent to zircon saturation, $Zr_{liquid}$ follows solid lines.

central Chilean arc where 10 Myr of plutonic activity culminated in the formation of the world's largest porphyry Cu deposit cluster at Río Blanco-Los Bronces[26,29,37].

## Results and discussion

### Early and low-temperature zircon saturation in super-wet magmas

We test the effect of increasing melt $H_2O$ on zircon saturation in crystallising andesitic magmas by combining the thermodynamic modelling software rhyolite-MELTS with zircon solubility models[38–40]. The aim is to explore the competing controls on Zr evolution in differentiating magmas: Zr increase due to its incompatible behaviour during major mineral crystallisation; and Zr decrease (or buffering) due to zircon saturation. These two controls operate in tandem and show a complex interplay. For example, $H_2O$ affects melt liquidus temperature and the rate of change of crystallinity and melt composition during cooling. To address this complexity our model builds isobaric temperature-crystallinity curves at initial water contents (wt.% $H_2O_i$) between 2 and 8 wt.% to cover the range inferred for arc magmas in the mid to upper crust (Fig. 1). Pressure is held at 400 MPa to allow higher water solubility (ca. 8 wt.% $H_2O$). We then use mineral-melt partitioning data (see "Methods" section) to model the increase of Zr with crystallisation and a zircon solubility model to predict the temperature at which zircon saturates[40]. We also test the effect of varying initial melt Zr (200 and 100 ppm, bracketing typical parent magma concentrations; Fig. 2).

Increasing the initial water content in the melt leads to zircon saturation at lower temperatures (Fig. 3A): when the initial Zr content of the magma ($Zr_i$) is fixed at 100 ppm, zircon saturates at 850 °C for 2 wt.% $H_2O_i$ and 790 °C at 8 wt.% $H_2O_i$ (Fig. 3B). When $Zr_i = 200$ ppm, these temperatures rise to 900 and 830 °C respectively. Furthermore, addition of $H_2O$ suppresses the increase in melt Zr with decreasing temperature, consistent with the near invariant Zr concentrations observed in the natural data from wet arc magmas (Fig. 2). The reasons for low-temperature zircon saturation in wet magmas in our models

are twofold. First, increasing initial melt water content of an andesitic magma from 2 to 8 wt.% $H_2O$ (at elevated pressure to ensure all $H_2O$ is dissolved) leads to a reduction in liquidus temperature from 1120 °C at 2 wt.% $H_2O$ to 1050 °C at 8 wt.% $H_2O$ and a corresponding displacement of temperature-melt fraction paths to lower temperatures (Fig. 3A). Consequently, at the point of zircon saturation, wetter magmas will contain significantly lower Zr (175 ppm at $H_2O_i = 2$ wt.% and 110 ppm at $H_2O_i = 8$ wt.%; Fig. 3B). Secondly, according to the zircon solubility model[40], water causes a modest increase in zircon solubility due to melt depolymerisation, leading to a shift of ca. −30 °C in zircon saturation temperatures from 2 to 8 wt.% $H_2O_i$ (Fig. 3B).

A key finding is that magmas with higher $H_2O$ will crystallise zircon at lower temperatures (Fig. 3). Such low-temperature zircon crystallisation may be recorded by low Ti concentrations because zircon Ti is known to be strongly temperature sensitive and is commonly used as a melt geothermometer[41,42]. Consequently, we investigate whether zircon Ti could be a fingerprint of particularly wet magmas because−unlike Zr systematics−it can be applied in the common case where a complete differentiation series of cogenetic magmas is not available.

### Tracking the transition from damp to super-wet magmatism

The strong control of magmatic $H_2O_i$ contents on zircon saturation, and the potential for it to be proxied by bulk-rock Zr and zircon Ti concentrations, provides a method to fingerprint super-wet arc magmas in natural rocks. We test this by tracking the long-term evolution of melt $H_2O$ in arc magmas from Central Chile between 18 and 4 Ma. This time window contains the transition from arc plutonism in the San Francisco Batholith (SFB) complex[43,44], to the formation of the world's largest porphyry Cu cluster at Rio Blanco-Los Bronces. This allows us to test whether a transition to super-wet magmatism is coincidental with porphyry Cu deposit formation. We report new zircon U-Pb geochronology and trace element geochemistry from the SFB and Los Bronces porphyry Cu intrusions, determined by laser ablation inductively-coupled-plasma mass spectrometry. Our primary focus here is on zircon Ti concentrations, as this allows zircon crystallisation

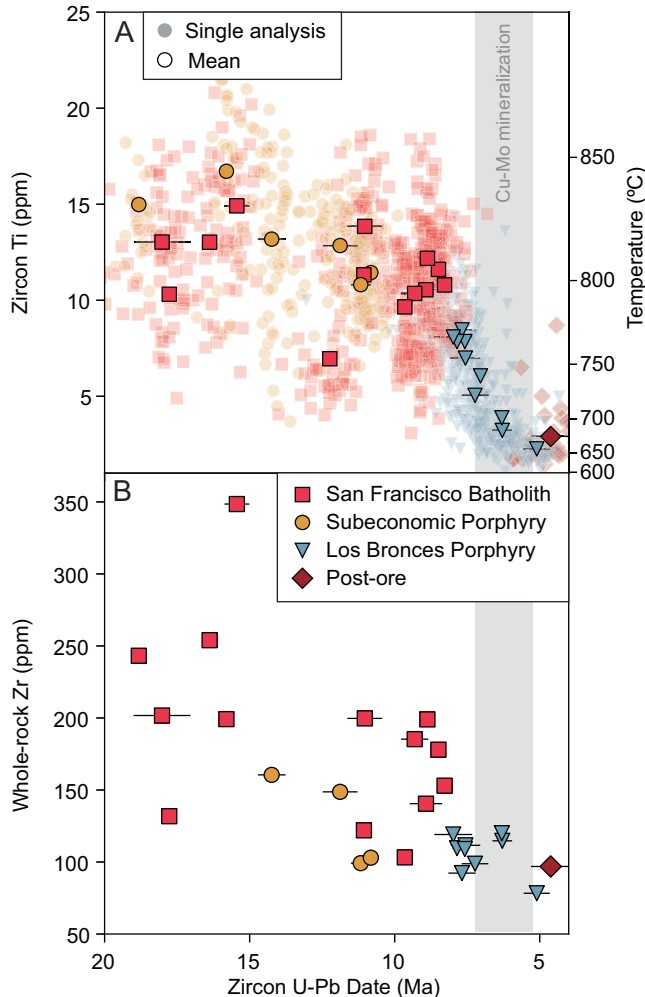

**Fig. 4 | Temporal evolution of Ti-in-zircon and bulk-rock Zr in Central Chile before and during porphyry Cu deposit formation.** Scatter plots showing **A** Ti-in-zircon concentrations and **B** whole-rock Zr over time from the San Francisco Batholith and Los Bronces porphyry Cu-Mo system in Central Chile[45]. For individual spot zircon Ti concentrations measured by LA-ICP-MS, the simultaneous spot $^{206}$Pb/$^{238}$U dates are shown (translucent symbols); solid symbols give the median Ti concentration and the weighted mean date for each sample. Vertical grey field in **A** and **B** indicates the inferred window for porphyry Cu mineralisation from the main cluster of molybdenite Re-Os dates from Los Bronces[51]. The right-hand axis in **A** shows the equivalent temperatures of zircon crystallisation determined using the Ti-in-zircon thermometer[42] assuming $a$TiO$_2$ = 0.7, $a$SiO$_2$ = 1.0 and $P$ = 400 MPa. For whole-rock Zr, the U-Pb dates are the zircon $^{206}$Pb/$^{238}$U weighted mean of the same sample. The error bars give the 2σ error on the weighted mean with a propagated 2% error to account for systematic uncertainty.

temperatures to be integrated with U-Pb dates to track the temporal evolution of melt temperature and thus, H$_2$O.

We find that rocks of the SFB (including sub-economic porphyries) contain zircon with variable Ti (5–23 ppm), whereas zircon from the Los Bronces porphyries has lower and more homogeneous Ti concentrations (2–9 ppm; Fig. 4A). Notably, the porphyry intrusions (after 8 Ma), which coincide with peak Cu-Mo mineralisation, display the lowest Ti-in zircon (<5 ppm), equivalent to the lowest zircon crystallisation temperatures. These lower zircon Ti concentrations cannot simply be attributed to more evolved magma compositions because they are also observed in basaltic-andesitic porphyry dykes at Los Bronces (see Supplementary Fig. 2). Based on our modelling, we interpret this change as a transition from dry-damp zircon

crystallisation to super-wet zircon crystallisation at ~8 Ma. Bulk-rock Zr in the older SFB samples is variable (130–350 ppm) with a gradual decline from 19 to ~8 Ma (Fig. 4B), consistent with damp arc magma evolution (2–4 wt.% H$_2$O; Fig. 2). This then transitions to lower and less variable bulk-rock Zr (80–130 ppm) in the Los Bronces porphyry Cu-associated intrusions (Fig. 4B), despite a range in SiO$_2$ contents (56–67 wt.% SiO$_2$) suggesting differentiation under super-wet conditions (>6–8 wt% H$_2$O; Fig. 2). The inferred increase in H$_2$O leading up to the formation of the Los Bronces porphyry cluster is contemporaneous with increased tectonic convergence and crustal thickening in Central Chile[45]. Elevated tectonic compression would promote magma storage at greater depths and for longer durations, potentially promoting the formation of super-wet melts during protracted magma differentiation[46].

## Low titanium-in-zircon as an indicator of super-wet magmas

A striking aspect of the Los Bronces zircon dataset is the very low and extremely restricted range of Ti concentrations of zircon from the economic porphyry Cu intrusions, which are almost entirely below 5 ppm in the samples linked to peak Cu-Mo mineralisation (Fig. 4A). In addition to temperature, Ti incorporation in zircon is dependent on the thermodynamic activity of titania ($a$TiO$_2$). Assuming $a$TiO$_2$ for arc-related magmas in the range of 0.4–0.9 (as measured from Fe-Ti oxide pairs; ref. 47 and Supplementary Fig. 4), these low Ti-in-zircon concentrations would equate to very low, near solidus, zircon crystallisation temperatures (749–678 °C; ref. 42). Our model shows that these temperatures could be achieved by very high-water contents (~8 wt.%) because zircon saturation occurs at temperatures below 790 °C in such melts (Fig. 3). A key question is whether these low, homogeneous zircon Ti concentrations can be achieved at upper crustal depths (200 MPa) where the maximum H$_2$O that can be dissolved in the melt is ~6 wt.%, or whether they require zircon crystallisation at greater depths in the crust where more water can be dissolved. We, therefore, integrated our zircon saturation model with Ti-in-zircon thermometry to evaluate the water contents and depths under which such low and homogeneous zircon Ti concentrations can be reproduced.

We first use mass balance to estimate the mass fraction of total zircon crystallised at a given temperature and melt fraction following zircon saturation (Fig. 5). Overall, we find that wet magmas crystallise a larger proportion of their zircon at lower temperatures, whereas dry-damp magmas crystallise zircon over a higher and wider range of temperatures. Because wet magmas undergo a greater proportion of their crystallisation at lower temperature (i.e., the slope of the crystallinity-temperature curve steepens; Fig. 3A), the peak of zircon crystallisation is shifted towards lower temperatures. Finally, increasing H$_2$O$_i$ causes zircon to appear as an earlier phase in the crystallisation sequence relative to other phases in the thermodynamic model (i.e., at higher melt fraction; Fig. 5A, B), because of the lower temperature of wet magmas and the very strong temperature dependence of zircon solubility (~30% crystals at 8 wt.% H$_2$O$_i$ compared to 55% crystals at 2 wt.% H$_2$O$_i$). Therefore, increasing H$_2$O$_i$ causes zircon crystallisation to take place over a longer crystallinity interval, albeit at relatively low and less variable temperatures.

We next model the distribution of Ti concentrations expected in zircon crystallised over a range of pressures and water contents. This integrates the mass fraction of zircon crystallised at a given temperature with models of rutile solubility (to estimate $a$TiO$_2$; ref. 48) in order to calculate the distribution of Ti-in-zircon concentrations expected at a given H$_2$O$_i$ (see "Methods" for further information). We find that increasing H$_2$O$_i$ causes lower melt $a$TiO$_2$ at zircon saturation due to the suppressed crystallisation of Ti-poor phases (e.g., plagioclase) and promoted crystallisation of Ti-rich minerals (e.g., magnetite and amphibole) at higher temperature, from $a$TiO$_2$ = 1.0 at 2 wt.% H$_2$O to $a$TiO$_2$ = 0.4 at 8 wt.% H$_2$O. The modelled Ti concentrations (Fig. 6)

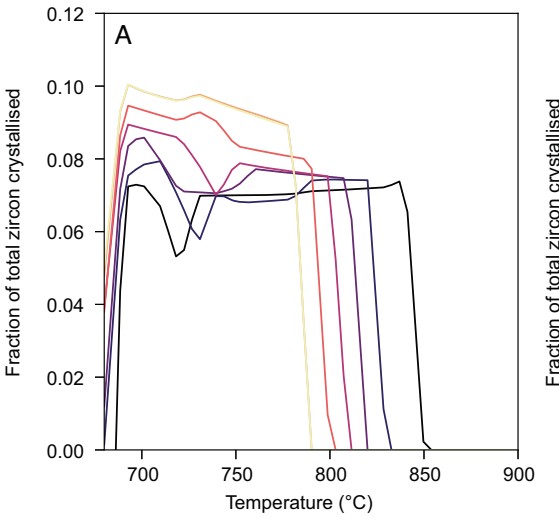
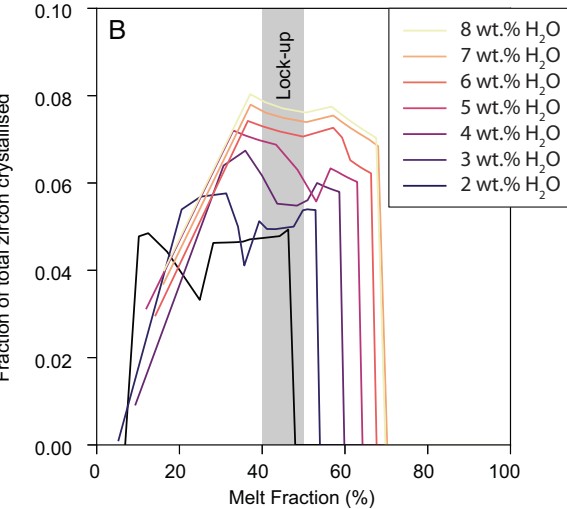

**Fig. 5 | Modelling the mass of zircon crystallised during magma crystallisation.** The fraction of total zircon crystallised in a magma as a function of **A** temperature and **B** melt fraction for $Zr_i = 100$ ppm and variable initial water content (coloured curves). The fraction of total zircon crystallised denotes the mass of zircon crystallised at each model increment divided by the total mass of zircon crystallised from the magma. The grey bar indicates the rheological lock-up when a melt-dominated mush transitions to a crystal-rich condition and magma becomes unextractable[77].

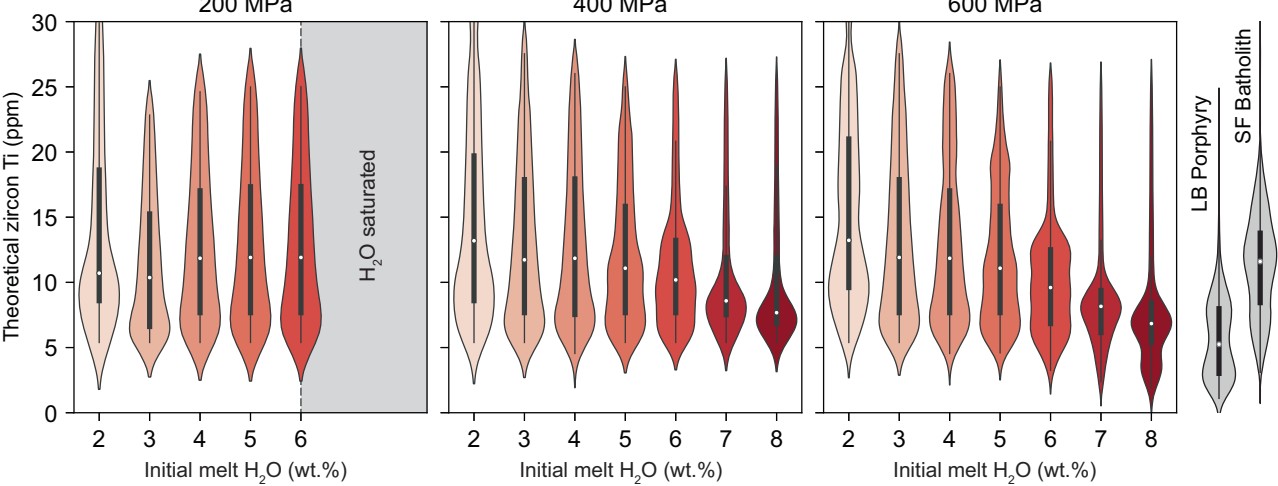

**Fig. 6 | Modelled Ti-in-zircon distributions as a function of $H_2O_i$ at three different pressures.** Violins show the density of data points and box-and-whisker symbols show interquartile range and median Ti concentrations. Grey violins and box plots on the right show, for comparative purposes, the distributions of Ti-in-zircon in samples from the Los Bronces (LB) porphyry intrusions and the San Francisco (SF) Batholith and uneconomic porphyries in the same district. The maximum solubility of $H_2O$ at 200 MPa is ~6 wt.%, hence higher $H_2O$ violins are not shown at this pressure.

show that increasing $H_2O_i$ at elevated pressure (400 MPa) causes a transition from higher and more variable zircon Ti (8–22 ppm interquartile range at 2 wt.% $H_2O_i$) to lower and more homogeneous zircon Ti (7–12 ppm interquartile range) at 8 wt.% $H_2O_i$ and an even more pronounced change at 600 MPa (Fig. 6). Conversely, the behaviour at 200 MPa is different due to a maximum $H_2O$ solubility of 6 wt.%. These results are broadly equivalent to zircon Ti data from the precursor San Francisco Batholith (low P and low $H_2O_i$) and from the mineralised Los Bronces porphyry intrusions (high P and high $H_2O_i$) in Central Chile (Fig. 6).

We conclude that high melt $H_2O$ is the primary control on the low and homogeneous Ti concentrations in zircon and attribute this to three contributory factors: (1) the lower temperatures of wet magmas due to the displacement of the liquidus and crystallisation to lower temperatures, thereby affecting the partitioning of Ti between melt and zircon; (2) the modest increase in zircon solubility with increasing

$H_2O$ due to melt depolymerisation and; (3) the lower melt $aTiO_2$ of wetter magmas at zircon saturation.

**Porphyry Cu deposit formation linked to super-wet magmas**

The temporal link between super-wet magmas and the development of Cu mineralisation in the Los Bronces porphyry system suggests that magmas with very high water contents could have a predisposition to form porphyry Cu deposits. This warrants a further comparison with literature data to examine whether low Ti zircon, and thus super-wet magmas, are a common feature in porphyry Cu-related magmatic systems. Porphyry Cu deposits are also commonly hosted by composite precursor plutons that are interpreted to have formed from typical arc magmas in terms of water contents (ca. 4 wt.%)[43,44]. If correct, these pre-mineralisation intrusions would be expected to have higher and more variable zircon Ti concentrations.

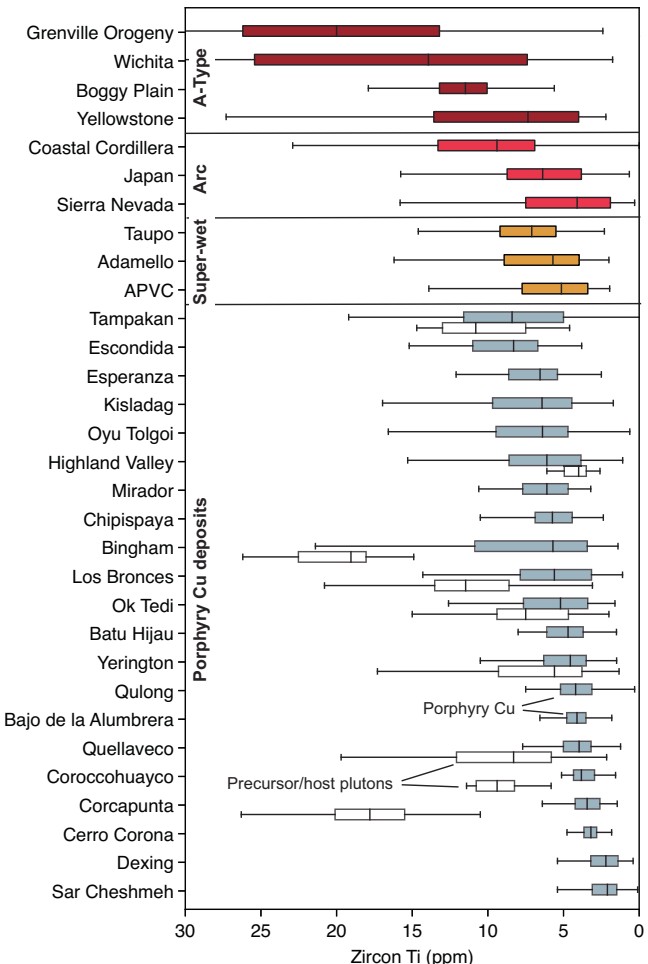

**Fig. 7 | Global comparison of zircon Ti concentrations across various rock types.** Box-and-whisker symbols show interquartile range and median Ti concentrations for each locality. Comparison includes zircon Ti concentrations measured in A-type systems ("hot" and "dry"), arc localities (Coastal Cordillera, Japan and Sierra Nevada) and localities where magmas are posited to be super-wet (Adamello, Altiplano Puna Volcanic Centre (APVC), Taupo Volcanic Zone) and porphyry Cu districts (blue = porphyry, white = precursor batholith where present/data available). Source data are provided as a Source Data file.

A compilation of zircon trace element compositions for 22 global porphyry copper deposits worldwide indicates that low and homogeneous zircon Ti (<10 ppm, 75th percentile) can be found in most of these systems (Fig. 7). Our modelling indicates that these zircon Ti distributions require melt water contents in excess of 6 wt.% (Fig. 6). Where zircon compositional data exist for precursor plutons in a porphyry Cu district, these have elevated Ti concentrations relative to porphyry Cu-associated zircon in all but one case (Highland Valley, Canada). We emphasise that our broad comparison groups all porphyry-related intrusions in a district together, whereas at Los Bronces we can resolve the specific porphyry dykes that have the closest association with Cu mineralisation in terms of lowest Ti-in-zircon.

Our comparison indicates that low zircon Ti concentrations are present in porphyry Cu deposits regardless of their size, grade or deposit type. Thus, super-wet arc magmatism exerts a first-order control on porphyry Cu mineralisation. Indeed, modelling studies have suggested that very high water contents may enable a magma to exsolve and outgas larger volumes of Cu-charged fluids[26,27,29]. This is also consistent with the observation that volcanoes with higher water

contents in their erupted magmas flux the largest masses of metals to the atmosphere[26]. Furthermore, because wet magmas have higher melt fractions at a given temperature, water saturation will occur at higher melt fractions where a larger volume of melt is available for scavenging of metals and the chlorine required for complexation and transport of metals in hydrothermal fluids[29]. If, as suggested by our models, these porphyry ore-forming, zircon-bearing magmas are stored at higher pressure (ca. 400 MPa, i.e., in the mid-crust), the insulating effect of the thicker overburden would extend the thermal lifetime of these systems[49]. This would allow multiple episodes of magma injection, storage and eventual fluid release over millions of years to form the superimposed generations of porphyry Cu intrusions that are typical of the largest mineralised districts[50,51].

### Underestimation of water contents in arc magmas

The link between super-wet arc magmas and porphyry Cu deposits raises the question of whether such magmas occur rarely and are unique to mineralised systems or are instead a common product of arc magmatism. To this end, we also compared zircon Ti contents between a number of arc magmatic systems without known association with porphyry systems, but where the existence of super-wet magmas has been proposed (Fig. 7), e.g. the Adamello Batholith (Italy), the Altiplano Puna Volcanic Complex (Bolivia), and the Taupo Volcanic Zone (New Zealand)[16,19]. Their zircon crystals also show relatively low and homogenous Ti contents (<10 ppm, 75th percentile) supporting their association with super-wet magmas. When compared broadly with other magmatic arcs (Fig. 7), the ranges in Ti values are similar to those observed in the Sierra Nevada (USA) and Japan, whilst the Coastal Cordillera (Chile) show higher and more variable zircon Ti (Fig. 7; ref. 52). We also compared the global porphyry data with zircon from A-type granites, which form under dry-damp and hot magmatic conditions. As expected, these show much higher and more variable Ti concentrations (interquartile range 8–26 ppm Ti; Fig. 7).

The occurrence of low zircon Ti concentrations in a number of arc rocks implies that deep, super-wet zircon crystallisation is not necessarily a rare process. This finding is at odds with the widely accepted range of $H_2O$ contents in arc magmas, typically inferred from melt inclusion data (0–6 wt.% $H_2O$; Fig. 1). Our finding implies that many arc magmas exceed this typical range (>6 wt.% $H_2O$) and therefore arc magma $H_2O$ contents may have been historically underestimated. This also raises questions over the mechanism by which such super-wet magmas ascend to the upper crust while avoiding decompression-related crystallisation and stalling at depth[53]. We tentatively propose that this is because their high water contents would initially reduce their viscosity and enable ascent so rapidly that initial dissolved water contents are retained, thus avoiding stalling and crystallisation at depth due to early $H_2O$ saturation[54]. This may provide a viable mechanism by which super-wet magmas can ascend to the upper crust and efficiently transfer their deep volatiles to upper crustal magmatic-hydrothermal systems, such as those associated with porphyry ore deposits.

### A deep locus of zircon crystallisation in arc magmas

The pressure at which the bulk of zircon crystallisation occurs in arc magmas is difficult to constrain but has significant implications because zircon is widely used to interpret magma storage durations, fluxes and volumes in trans-crustal magmatic systems[55,56]. The strong effect of increasing melt $H_2O$ on promoting early zircon saturation in arc magmas (Fig. 5B) would influence the initial depth at which zircon begins to crystallise. Zircon formed during the crystallisation of super-wet magmas may therefore represent cargo from multiple crustal depths, with good preservation due to zircon's refractory nature and low dissolution rates[57]. Enhanced zircon stability by high melt $H_2O$ could explain the protracted timescales of zircon crystallisation documented by high-precision U-Pb dating in porphyry Cu deposits

such as at Bingham Canyon (~650 kyr; ref. [58]), Batu Hijau (~300 kyr; ref. [59]), Bajo de la Alumbrera (~200 kyr; ref. [60]), Coroccohuayco (>300 kyr; ref. [61]), Koloula (>100 kyr; ref. [62]) and Ok Tedi (~200 kyr; ref. [63]). Such timescales are also found in a number of other hydrous magmatic systems such as the Adamello Batholith (Italian Alps) and the Fish Canyon Tuff (USA)[64,65]. This suggests that >100 kyr durations of zircon crystallisation in arc magmas, which are typically interpreted as storage durations in the upper crust, may instead represent a protracted, deeper magma history. By contrast, much shorter timescales of zircon crystallisation have been recorded in dry (A-type) magmatic systems, such as the Kilgore Tuff (Yellowstone, USA) where high-precision U-Pb dates of multiple zircon grains are statistically indistinguishable (<10,000 years of zircon crystallisation[66]).

Our model suggests that magmas forming porphyry Cu deposits are stored at mid-crustal depths (>400 MPa, ~15 km), as opposed to in the upper crust[67]. This does not preclude subsequent storage of these magmas for shorter timescales at shallower depths (i.e., at 5–10 km depths[68]) where the volume of crystallised zircon might be subordinate. Mid-crustal depths of magma storage and zircon crystallisation are in accordance with the large volume of intermediate-felsic magmas inferred at such depths from geophysical surveys in mature magmatic arcs[16,69] and from tilted crustal sections such as in the Sierra Nevada (felsic rocks up to 35–40 km depths[70]) and the Famatinian arc (intermediate to felsic rocks dominate until 30 km depths[71]).

In conclusion, we have shown that zircon saturation in arc magmas is strongly sensitive to melt $H_2O$ contents. We find that low, homogeneous zircon Ti concentrations (<10 ppm, 75th percentile) reflect zircon crystallisation from low-temperature, super-wet magmas (>6 wt.% $H_2O$) in mid-crustal (or deeper) magma reservoirs. A transition from high and variable to low and homogeneous Ti zircon compositions coincides with the formation of the world's largest Cu resource in Central Chile. Deep, super-wet zircon crystallisation appears to be a common feature of porphyry Cu deposits worldwide, hinting that super-wet magmas are fundamental vehicles for delivering large metal fluxes towards Earth's surface.

## Methods

### Trace element analysis of zircon

Rocks were crushed and sieved to 500 μm, panned for heavy minerals and then magnetic minerals were removed using a neodymium magnet. Zircon crystals were hand-picked from the mineral separates, mounted in epoxy and polished to expose the crystal cores. The zircon crystals were then imaged by scanning electron microscope-cathodoluminescence (SEM-CL) to allow textural examination. Trace element compositions and U-Pb dating of zircon were performed at the Imaging and Analysis Centre at the Natural History Museum (London, UK) using an Agilent 7700x quadrupole ICP-MS coupled to an ESI New Wave Research NWR193 excimer laser. A complete description of the LA-ICP-MS method can be found in Supplementary Data 1. The laser was operated using a 5 Hz repetition rate, a fluence of 3.5 J cm$^{-2}$ and a spot size of 30 μm. Laser spots were selected to cover the range of textural zones exhibited by the zircon crystals (e.g., cores and rims). NIST-610 was used as the primary standard for trace elements and 91500 was used as secondary reference material. For U-Pb geochronology, GJ-1 was used as the primary standard and AUSZ-7 and Rak-17 standards were used to monitor consistency and accuracy in the U-Pb data. Zircon trace element data, U-Pb ages and analyses of standards are provided in Supplementary Data 2–5.

### Modelling of zircon saturation as function of initial melt $H_2O$

**Rhyolite-MELTS setup.** Temperature-crystallinity curves for different initial water contents in intermediate-felsic arc magmas were modelled using rhyolite-MELTS[38,39]. Modelling was carried out using the *thermoengine* module for Python in the ENKI server (https://enki-portal.gitlab.io/ThermoEngine/index.html). An intermediate bulk starting

composition was used from the Adamello Batholith (Northern Italy) because this is a calc-alkaline subduction-related andesite and it is not hydrothermally altered[72]. The system was equilibrated at a starting temperature of 1100 °C to ensure 100% liquid, and the temperature was decreased in 10 °C increments to a near solidus temperature of 680 °C to ensure full crystallisation. These models were performed using the *equilibrate_tp* function of *thermoengine* assuming fractional crystallisation. Minerals that are not expected in calc-alkaline magmas were suppressed to simplify the modelling. The melt composition (calculated on an anhydrous basis), phase assemblage and melt fraction were extracted at each model increment. These models were repeated for a range of water contents (2–8 wt.% $H_2O$) and pressure (200–600 MPa). The code used to run these simulations is archived at the point of submission at (https://doi.org/10.5281/zenodo.10557188).

**Calculating melt Zr concentrations.** Rhyolite-MELTS cannot calculate melt Zr ($Zr_{melt}$) since it does not currently implement a thermodynamic model for trace elements. Therefore, to model $Zr_{melt}$ we use simple Rayleigh fractional crystallisation:

$$Zr_{melt} = Zr_0 F^{D-1} \tag{1}$$

Where $Zr_0$ is the initial Zr concentration of the melt, F is the melt fraction and D is the mineral-melt bulk partition coefficient for Zr. The bulk partition coefficient of Zr is likely to be very low due to the general incompatibility of Zr in the major mineral phases found in porphyry rocks (feldspar, amphibole, biotite and magnetite). For plagioclase and magnetite, the partition coefficients are very close to 0, whereas for amphibole ($D_{Zr} = 0.1–2.4$) and biotite ($D_{Zr} = 0.42$) the partition coefficients can be higher (particularly at low temperature). In order to track $Zr_{melt}$, we follow the approach of previous work[72] where we use compilations of mineral-melt Zr partitioning data to constrain the bulk mineral-melt partition coefficient for Zr (Supplementary Fig. 1) and the mineral fractions outputted from rhyolite-MELTS. We note that amphibole was present in some runs but is generally underestimated due to the simplification of the amphibole model by rhyolite-MELTS, and hence bulk $D_{Zr}$ should be treated as a minimum. Data for $D_{Zr}$ as a function of temperature were compiled for amphibole, clinopyroxene, plagioclase, biotite and ilmenite (Supplementary Fig. S1). Due to a paucity of literature data (particularly at lower temperatures), $D_{Zr}$ was estimated for orthopyroxene and spinels (magnetite) by taking a third of the values for clinopyroxene and ilmenite respectively, based on a comparison of experimental partitioning data for these minerals[72]. Because we expect $D_{Zr}$ to increase with decreasing temperature, we fit a 3rd order polynomial equation to temperature and $D_{Zr}$ for each mineral phase (Supplementary Fig. 1). We fit this solely to temperature rather than $SiO_2$[72] because temperature is an independent variable and melt $SiO_2$ has uncertainty since the rhyolite-MELTS models do not crystallise expected amphibole abundances.

**Zircon saturation modelling.** At a given temperature step, the Zr concentration required for zircon saturation can be calculated using the empirical calibration of Crisp and Berry (ref. [40]) which is based on experimental studies. The model is dependent on the temperature ($T$) in K, pressure ($P$) in GPa, melt optical basicity ($\Lambda$ – a compositional parameter) and the mole fraction of water in the melt ($xH_2O$). For a given model step $i$, the Zr required for zircon saturation is given by:

$$\log Zr_{sat(i)} = 0.96(\pm 0.05) - \frac{5790(\pm 95)}{T_i} - 1.28(\pm 0.08)P$$
$$+ 12.39(\pm 0.35)\Lambda_i + 0.83(\pm 0.09)(xH_2O)_i + 2.06(\pm 0.16)P\Lambda_i \tag{2}$$

Where Zr is in ppm and uncertainties in parentheses are 1σ. This equation is also frequently rearranged to calculate minimum

temperatures of zircon-saturated magmas[73]. The optical basicity at a given model step can be calculated from the MELTS output of liquid composition[74]. A limitation of this approach is that rhyolite-MELTS is not optimised for wet intermediate-felsic magma compositions due to the lack of an appropriate thermodynamic model for hydrous mafic silicates such as amphibole[38]. As such, at higher degrees of crystallinity our residual melt compositions show an inflection with differentiation where $SiO_2$ decreases (Supplementary Fig. 6). This inflection is not consistent with liquid compositions from experiments (Supplementary Figs. 6 and 7) or natural glasses and would lead to inaccuracies in our zircon saturation predictions. In general, we find that the MELTS liquid $SiO_2$ diverges from experimental data below 850 °C. Therefore, we perform a polynomial regression on optical basicity (which is a function of the major element composition of the melt) versus temperature from the liquidus to 850 °C and use this to correct for the inflection of melt composition with lower temperature. A 2nd order polynomial was selected based on the best fit to experimental data (Supplementary Figs. 6 and 7).

**Calculating theoretical zircon Ti concentrations.** Once the melt achieves zircon saturation (i.e., where $Zr_{melt} > Zr_{sat}$), we use mass balance to calculate the fraction of zircon crystallised at each model step. This produces a distribution of the relative mass of zircon that would be crystallised at a given temperature or melt fraction (Fig. 5). The relative mass of zircon at a given temperature, as calculated from our rhyolite-MELTS and zircon saturation modelling can then be used to calculate a range of theoretical Ti concentrations in crystallised zircon using the Ti-in-zircon thermometer[42]:

$$\log(\text{Ti}, f) = 5.84(\pm 0.07) - \frac{4800(\pm 86)}{T} - 0.12(\pm 0.01)P$$
$$- 0.0056(\pm 0.0015)P^3 - \log\left(a_{SiO_2}\right)f + \log(a_{TiO_2}) \quad (3)$$

Where Ti is the zircon Ti concentration in ppm, $T$ is temperature in K, $P$ is pressure in GPa and $f$ is the fraction of Ti on the zircon tetrahedral site which is calculated as[42]:

$$f = \frac{1}{1 + 10^{(-0.77(\pm 0.05)P + 3.37(\pm 0.13))}} \quad (4)$$

The numbers in parentheses are 1σ errors. The chemical activities ($a_{TiO_2}$ and $a_{SiO_2}$) can be calculated using the affinity ($A$) of $TiO_2$ and $SiO_2$ in the melt which are output from rhyolite-MELTS, for example:

$$a_{TiO_2} = e^{\frac{-A_{TiO_2}}{RT}} \quad (5)$$

Where $R$ is the gas constant. In general, MELTS predicts ranges of 0.3–0.6 and 0.8–1.0 for $a_{TiO_2}$ and $a_{SiO_2}$ at the time of zircon saturation. This is in general accordance with values predicted for I-type magmas from other attempts to reconstruct $a_{TiO_2}$ and $a_{SiO_2}$ using thermodynamic modelling software[75] although the values for $a_{TiO_2}$ are generally lower than those predicted by coexisting Fe-Ti oxides in volcanic rocks (0.4–0.9[47]; Supplementary Fig. 4). Many of these higher (>0.6) $a_{TiO_2}$ values are found in calc-alkaline andesites-dacites such as Pinatubo and Mt St Helens[47], suggesting that rhyolite-MELTS may underestimate $a_{TiO_2}$[76]. We also find that at low temperatures (<800 °C) the values of $a_{TiO_2}$ can unrealistically collapse to zero as $TiO_2$ reaches very low concentrations in the melt. Therefore, we calculate melt $a_{TiO_2}$ using the recent rutile solubility model of Borisov and Aranovich[48] from the melt compositions derived by our MELTS simulations to estimate $a_{TiO_2}$ at each model step (Supplementary Fig. 4B). In general, we find the Borisov and Aranovich model produces higher $a_{TiO_2}$ compared to rhyolite-MELTS (0.4–1.0) and even predicts

rutile saturation in some low-$H_2O$ models. (Supplementary Fig. 4). These higher $a_{TiO_2}$ values are in better agreement with the ranges predicted from Fe-Ti oxides in calc-alkaline lavas. We therefore implemented the method of Borisov and Aranovich to model zircon Ti concentrations.

**Ethics and inclusion**
This study benefited from collaboration and knowledge from local geologists at Anglo American. Local assistance and knowledge have been rightfully acknowledged/cited.

## Data availability
Zircon U-Pb and trace element data generated in this study are provided in Supplementary Data and are deposited in the publicly available repository: https://doi.org/10.5281/zenodo.12794414. Source data are provided with this paper.

## Code availability
The Python code for the modelling of zircon saturation and zircon Ti concentrations is publicly available and deposited at: https://doi.org/10.5281/zenodo.12794414.

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

## Acknowledgements

C.N. was funded by an ETH Zurich Postdoctoral Fellowship (22-1 FEL-21). C.N., J.B., S.J.E.L., J.J.W., and Y.B. acknowledge funding from Natural Environment Research Council grant (NE/P017452/1) "From arc magmas to ores (FAMOS): a mineral systems approach". J.B. acknowledges support from a Royal Society Research professorship (RP\R1\201048). CCM was supported by the Swiss National Science Foundation (grant 200021_212892). Field sampling and logistical support from Anglo American and the Anglo Chile team, in particular Sebastien Ramirez, Ricardo Boric, Pablo Villegas and the technical staff is greatly appreciated. We thank Callum Hatch for sample preparation and Dawid Szymanowski and Christoph Heinrich for insightful discussions. This paper benefited from constructive input from Marco Fiorentini, Yong-Jun Lu, and Calvin Miller.

## Author contributions

Conceptualisation: C.N., J.B. Methodology: C.N., S.J.E.L., Y.B., L.T., C.C.M. Investigation: C.N., J.B., S.J.E.L., J.J.W., M.L., Y.B., L.T., C.C.M. Visualisation: C.N. Writing—original draft: C.N. Writing—review & editing: C.N., J.B., J.J.W., M.L., Y.B., L.T., C.C.M.

## Competing interests
