## [Peer Review File · Nature Communications]

REVIEWER COMMENTS

Reviewer #1 (Remarks to the Author):

Review of manuscript "A zircon case for super-wet arc magmas" by Nathwani and co-authors for Nature Communications.

Dear authors and editor,

I have read with great pleasure this very interesting manuscript by Nathwani and co-authors. However, although the data and observations are interesting, I do think that the outcomes do not warrant publication in Nature Communications at this stage. The main problem in my opinion lies in how the narrative is structured, and how the topic is discussed. My criticism aims to be constructive here as I do believe that the data and associated modelling would be of great value to the scientific literature. However, they are currently written in such a way that is typical for a more discipline-specific journal, where the audience would be more prepared to understand the significance of the results.

First, I personally think that the text needs a significant rewrite from a syntax point of view. The paragraphs and sentences are way too long, and there are significant repetitions and inconsistencies. It is very wordy in places, and it is just hard to follow. I wonder whether the manuscript lacked a final stage of polishing, which would have significantly enhanced the message to be conveyed.

I also find that the structure of the manuscript is not conducive to a clear understanding of the key problems that are being addressed and their proposed resolution. In fact, results and discussion are mingled (they even sit in the very same section), and there is a lack of the clear explanation of the significance of the data that are being presented. The flavour of the discussion ranges from the link between volatile content and metal fertility, but then at the very end the take home message is centred on volatile flux through time. The result is that I finished reading the manuscript without really knowing what the key message was.

Once again, I hope that the authors understand that I am not being dismissive here. There is clearly a great deal of work being presented in the manuscript, and a number of excellent working hypotheses that are being put forward. But it is the way this is done that does not work for me. And if it was difficult to follow the argument for a person who is somehow verse with the topic, or at least aware of the key problems being discussed, I can only imagine that it would be even more difficult for a geoscientist with a different background.

The last point I would like to make, but not in order of importance, is that from the point of view of the message that is being conveyed there is a significant problem. The perception is that the authors are making a case that there is a spatial and genetic link between volatile contact of magmas and porphyry copper mineralisation. I think that we can all agree that this is fairly well established and that there is nothing new here. The point that should be made, and around which the entire novelty and excitement of this work could be, is whether there is a link between the genesis of super-wet magmas and the formation of world-class camps and districts. In other words, it is not about the genesis of porphyry copper mineralisation, which can happen in many different areas in arc environments, but it is about the

genesis of high-grade systems, and or mineralised camps. This message does not come across at all and should be discussed further.

I have added a few points on the annotated draft. Whereas I cannot recommend acceptance at this stage, I would like to commend the authors for a fantastic set of data, which will be hopefully rewritten in such a way that the key messages can be conveyed to the scientific literature.

Reviewer #2 (Remarks to the Author):

Review letter of “A zircon case for super-wet arc magmas” by Nathwani and coauthors.

High magmatic water contents are prerequisite for the formation of economically important magmatic-hydrothermal systems, such as porphyry Cu deposits. However, the direct evidence for such super-wet magmas is rarely preserved due to degassing at shallow crustal levels. The authors innovatively used thermodynamic modelling on zircon, a mineral that survives various lower-to-upper crustal geological processes, to quantify the role of water on zircon saturation. Through comprehensive modelling, they demonstrated that increasing water content in magmas leads to low and narrow ranges of zircon crystallization temperatures. The authors use zircon Ti concentrations as a proxy for magma water content and provide compelling evidence for the prevalence of super-wet magmas in porphyry Cu systems. The observed low and homogeneous Ti concentrations in zircons from porphyry Cu deposits suggest a link between super-wet magmatism and mineralization processes, with important implications for ore deposit formation.

The manuscript is well-written with conclusions supported by robust modelling and provides a significant contribution to our understanding of arc magmatism and associated mineralization processes.

Overall, I believe this manuscript represents a strong candidate for publication in Nature Communications. However, I recommend minor revisions to clarify certain points and strengthen the manuscript's overall coherence. Detailed comments and suggestions for improvement are provided below.

1. Abstract: it is worth mentioning how to use zircon to pinpoint super-wet magmas, such as low and homogeneous zircon Ti (< 12 ppm, 75th percentile) concentrations, and how this new understanding can help exploration of porphyry Cu deposits to meet the increasing demands of critical metals as the world transitions to green energy economy.

2. The definition of super-wet magmas is inconsistent throughout the MS. In the abstract, it refers to >6 wt% H₂O, but in the main text it is >8 wt% H₂O (e.g. Line 46). Please ensure a consistent definition is used throughout.

3. Line 124, change “lower” to “higher” temperatures...

4. Line 126, delete “observed with differentiation” as it is duplicate.
5. Line 130, it is not clear how Fig. 3A shows “systematic reduction in liquidus temperature from 1120 C to 1000 C”. Perhaps rephrase it or label the liquidus temperature on Fig. 3A.
6. Line 135-136, the statement of “with a shift of approximately +30 C in zircon saturation temperatures from 2 to 8 wt% H₂O_i (Fig. 3B)” is not consistent with the figure. Fig. 3B shows that zircon saturates at 850°C for 2 wt.% H₂O_i and 790°C at 8 wt.% H₂O_i when the initial Zr content of the magma is fixed at 100 ppm. So it is a shift of -60 C (not + 30 C) from 2 to 8 wt% H₂O_i.
7. Line 138, unclear what “high H₂O will do so at lower temperatures...” mean. Please rephrase.
8. Line 181, please change “<780 C” to “<790 C” to be consistent with Line 122.
9. Line 193, “the slope of the crystallinity-temperature curve steepens; Fig. 3A”. It appears Fig. 5A rather than 3A should be cited here.
10. Line 194-196, “Increasing H₂O_i causes zircon to appear as an earlier phase in the crystallisation sequence relative to other phases in the thermodynamic model (i.e., at lower melt fraction; Fig. 2B and D”. The citation of Fig. 2B and D is apparently incorrect. Are you referring to Fig. 5B? If so, lower melt fraction should be higher melt fraction if zircon crystallized earlier than other phases.
11. Line 522, NIST-612 data are missing in Supplementary Table 9.
12. Line 523, delete “and GJ-1” after Rak
13. Supplementary Table 1, the ablation pit depth and spot diameter should be in μm rather than mm.
14. Supplementary Table 2, please add the ²³⁸U/²⁰⁶Pb ratios and uncertainties so that Tera-Wasserburg Concordia can be plotted.

Reviewer #3 (Remarks to the Author):

Your ms is an excellent contribution to very important, timely issues. I believe that it can be made even better with some rather simple, minor tuning, as I'll describe below and in comments and a very few edits in the text.

Strengths are evident: you simultaneously address Big Geoscience Questions, both applied: generation of critical metal (Cu + others) deposits – and fundamental: arc magmatism, role of H₂O, construction of ~permanent crust. And you do so carefully and convincingly, with a novel, ambitious but well-thought-out combination of modelling techniques and regional and global data sets. I expected to be skeptical, at

least in part, but I find the discussion of methods and interpretations to be quite satisfying. And the paper is very well-written and illustrated and has good supporting documentation.

As I said above, I do have some suggestions, mostly regarding clarity and/or expanding on a couple of things. I'll start with a couple of very minor suggestions about supplementary material (which is for the most part exemplary):

◇ Fig S2: Are bold symbols sample means and muted symbols individual measurements? I suspect so, but make this clear in caption.

◇Supplementary Tables: give them abbreviated names in the tabs, to make it easier for readers to find data of interest.

◇Sup. Table 6 ("Whole-rock Zr compilation for different melt H₂O contents') is, I believe, referred to as Sup. Table 7 ("Compilation of whole-rock compositions from porphyry Cu districts") in Figure 2 caption.

I have inserted a number of comments – some of which might lead to minor modifications, others are thoughts that come to my mind while reading your text. Also a very few suggested small edits (wording, punctuation).

Below I expand on four comments, keyed to the text:

Line 206, Note #1: I would recommend expanding and clarifying a bit -- what is "Ti-free?" All minerals have some Ti, though negligible in qtz & feldspar (and zircon). But mafic silicates – especially hydrous mafic silicates- have a fair amount, many accessories (oxides, titanite, chevkinite) have a lot. BUT R-MELTS doesn't model these minerals well, if at all. How do you assess the role of significantly Ti-bearing minerals ("non-Ti-free")? You cover this pretty well in your Methods section, but when you address the Ti approach in the main text, I think it would be to expand a bit on how you've dealt with it or at least sent reader to Methods.

Line 528, Note #2: It seems like a stretch to say that modelling was done simply using Rhyolite-MELTS. What you describe in this section is not modelling of zircon saturation – you don't mention zircon at all except in this sentence. Results of what you describe here can (and are) used as input into zircon saturation modelling, and you present this approach in the following sections.

Line 649 (Fig. 2), Note #3: Your SiO₂-Zr data are not for magmas, at least not strictly – they are whole rock compositions, which, especially for the intrusive rocks, could differ significantly from magma compositions. And the H₂O estimates in Sup Table 6 (not 7) at least mostly appear to be based on melt (not magma). I think that this is a very informative figure, but I would augment the caption to make clearer what the data (SiO₂-Zr, H₂O) are based on.

Line 689 (Fig. 5), Note #4: I think that this figure needs to be clarified (caption and/or y-axis title) I think that Y values are confusing: what does "Fraction of total zircon crystallized" mean? Taken literally, to me at least, that value should increase monotonically down-T (no valleys in the trend) and reach 1.0 at the solidus. Is the Y value actually the fraction of zircon crystalized within a finite T range? Or what?

All in all, an excellent, thought-provoking paper! I enjoyed reading it and look forward to seeing it published!

Reviewer #1 (Remarks to the Author):

Review of manuscript "A zircon case for super-wet arc magmas" by Nathwani and co-authors for Nature Communications.

Dear authors and editor,

I have read with great pleasure this very interesting manuscript by Nathwani and co-authors. However, although the data and observations are interesting, I do think that the outcomes do not warrant publication in Nature Communications at this stage. The main problem in my opinion lies in how the narrative is structured, and how the topic is discussed. My criticism aims to be constructive here as I do believe that the data and associated modelling would be of great value to the scientific literature. However, they are currently written in such a way that is typical for a more discipline-specific journal, where the audience would be more prepared to understand the significance of the results.

First, I personally think that the text needs a significant rewrite from a syntax point of view. The paragraphs and sentences are way too long, and there are significant repetitions and inconsistencies. It is very wordy in places, and it is just hard to follow. I wonder whether the manuscript lacked a final stage of polishing, which would have significantly enhanced the message to be conveyed.

I also find that the structure of the manuscript is not conducive to a clear understanding of the key problems that are being addressed and their proposed resolution. In fact, results and discussion are mingled (they even sit in the very same section), and there is a lack of the clear explanation of the significance of the data that are being presented. The flavour of the discussion ranges from the link between volatile content and metal fertility, but then at the very end the take home message is centred on volatile flux through time. The result is that I finished reading the manuscript without really knowing what the key message was.

Once again, I hope that the authors understand that I am not being dismissive here. There is clearly a great deal of work being presented in the manuscript, and a number of excellent working hypotheses that are being put forward. But it is the way this is done that does not work for me. And if it was difficult to follow the argument for a person who is somehow versed with the topic, or at least aware of the key problems being discussed, I can only imagine that it would be even more difficult for a geoscientist with a different background.

The last point I would like to make, but not in order of importance, is that from the point of view of the message that is being conveyed there is a significant problem. The perception is that the authors are making a case that there is a spatial and genetic link between volatile contact of magmas and porphyry copper mineralisation. I think that we can all agree that this is fairly well established and that there is nothing new here. The point that should be made, and around which the entire novelty and excitement of this

work could be, is whether there is a link between the genesis of super-wet magmas and the formation of world-class camps and districts. In other words, it is not about the genesis of porphyry copper mineralisation, which can happen in many different areas in arc environments, but it is about the genesis of high-grade systems, and or mineralised camps. This message does not come across at all and should be discussed further.

I have added a few points on the annotated draft. Whereas I cannot recommend acceptance at this stage, I would like to commend the authors for a fantastic set of data, which will be hopefully rewritten in such a way that the key messages can be conveyed to the scientific literature.

We thank the reviewer for the constructive review on our manuscript. We are pleased to see the positive feedback on our data and modelling and agree that we could present this in a better way. We have now made major changes to our manuscript to address the concerns raised regarding the structure, clarity and implications of our study.

We have restructured things significantly to make the discussion clearer. It is not simple in our case to separate results and discussion fully, because for this journal we feel it would be better to have a dialogue where results are discussed in a narrative which guides the reader through the scientific line of thinking. However, we do agree, that the very large “results and discussion” section makes it challenging for the reader to navigate themselves. We have thus moved the last few subsections into a final section called “implications” which provides a clear explanation of the significance of what we have presented. We have removed the final discussion about volatile fluxes through time, we agree with the reviewer it is distracting and very speculative. We believe there are three take home messages of our paper:

1. Magmas forming porphyry Cu deposits are super-wet and we have placed a minimum constrain on their H₂O contents >6 wt.%
2. Super-wet magmas are not exclusive to PCDs suggesting that water contents of arc magmas have been typically underestimated
3. Zircons in arc magmas record, in many cases, a deeper magmatic history than previously interpreted with significant implications for magma storage depths and durations.

Each of these points is given a separate subheading in the “implications” section. We also provide a final paragraph which summarises the key conclusions of the paper so that the reader can take away these messages. We have also had multiple iterations reviews by co-authors to refine the manuscript and feel it is now in better shape. This also includes splitting paragraphs and sentences, where possible, to be more succinct and less wordy.

Regarding the novelty of our study, we agree that the link between volatile content of magmas and porphyry copper deposit formation is relatively well known. However, it is not widely known that these magmas are “super-wet” (i.e. form from magmas with >6 wt.% H₂O). For example, Richards (2011) states that H₂O contents of >4 wt.% in intermediate-felsic magmas are required. The classic review paper of Sillitoe (2010) also claims that an H₂O content of ~>4 wt.% is sufficient. More recently, Rezeau & Jagoutz (2020) show that the initial H₂O content of mafic arc magmas is what is important (i.e > 2 wt. H₂O) but not necessarily reaching super-wet conditions. Chiaradia (2020) also showed a similar point that arc basalts with normal H₂O contents are the most porphyry fertile but do not suggest these evolve to being super-wet. On the other hand, Loucks & Fiorentini (2023) suggest based on Zr concentrations of porphyry magmas that these have > 9 wt.% H₂O, which is certainly a new idea with support but not (yet) a long-standing state-of-the-art.

The previously interpreted link between wet magmas and porphyry copper deposits was relatively circumstantial rather than shown directly. For example, the link with high Sr/Y magmas is known but this has not permitted quantification of the H₂O contents of these magmas. We highlight that our study quantified this water content using natural data from the world’s largest deposit, and a large number of published data and petrological modelling. We do not feel that we can make the direct link between variation in deposit size/grade and the water content of magmas. In Figure 7, the districts with the lowest zircon Ti contents are not necessarily the largest/highest grade, e.g. Corcapunta, Coroccohuayco and Bajo de la Alumbrera. The fact that such an observation is made across different deposit types such as Cu-Mo porphyries (e.g. Los Bronces, Quellaveco) and Au-rich porphyries (Dexing, Bajo de la Alumbrera) indicates to us that super-wet magmas exert a first-order control on forming a porphyry Cu deposit, whilst the tonnage, grade and deposit type is dictated by other factors.

We have addressed the comments made by the reviewer on the PDF directly below.

L33-36: I assume this is the observation from one of the experiments? Maybe reference.

Yes, we have now added a reference to the three papers (Nandedkar et al. 2014, Melekhova et al. 2015 and Chang & Audetat 2023) that are used for the experimental data in Figure 1. See L39 of the revised ms.

L36: New paragraph after minerals.

We have added a new paragraph here as suggested.

L46: In the introduction it is > 6%

This was also pointed out by the other reviewers, and we have corrected to >6%

L48: Also evidence from chemistry and plagioclase suppression?

Yes, this is probably also a fair line of evidence, for example high Sr/Y ratios of magmas, although difficult to quantify an H₂O value, likely points towards particularly high H₂O contents. We have added this into the sentence, see L49-53 of the revised ms.

There is a growing body of evidence for the formation of ‘super-wet’ magmas at depth in arcs (>6 wt.% H₂O), such as from the high electrical conductivity of mid-crustal magma reservoirs¹³, high volatile contents in lower-crustal cumulates¹⁴ **suppression of plagioclase crystallization in some magmas**¹ and comparisons of cumulate mineral assemblages with experimental observations¹⁵⁻¹⁷.

L54: Super-wet magmas? Please be consistent with the terminology

We have edited this to now read “super-wet magmas” to keep consistent nomenclature

L61: Would extremely high water contents affect oxidation rates? Cf. Loucks and Fiorentini 2024 EPSL.

Indeed, it is possible that high H₂O will increase fO₂ – see also for example the upper crustal degassing experimental study of Humphreys et al. (2015; Journal of Petrology). However, we prefer not to get into the topic of deep crustal redox as it may also depend on other factors (e.g. the crust that is melted/assimilated) and do not want to deter from the key points we focus on later in the discussion.

L77-85: It would be good to provide a reference here.

Yes good point. We have added references to Keller et al. (2015) and Lee & Bachmann (2014) to this paragraph for the sentences about MORB and arc whole-rock Zr contents respectively.

L88-90: This sentence needs help

This sentence indeed did not make full sense and we have rewritten it as follows (see L94-96 in the revised ms):

Considering arc lavas where H₂O has previously been constrained, we find higher dissolved H₂O (4-8 wt.%) causes the increase in Zr with differentiation to become even further suppressed compared to arc lavas with lower H₂O (2-4 wt.%; Fig. 2).

L151-152: The link between super-wet conditions and fertility to generate giant mineralising systems is not emphasised enough at the beginning.

We agree that this is a key topic of our paper. We tried not to focus the introduction too strongly on mineralised systems because we wanted to keep the paper to a broad readership (i.e. a broad introduction with relevance to all arc magmas), but with the specification that super-wet conditions will have importance to ore deposit formation. We indeed note that water has been long known as pre-requisite in magmas that form porphyry Cu deposits but that exact quantification of this has been speculative. Furthermore, the previously proposed H₂O contents of magmas that form porphyry Cu deposits were not all necessarily super-wet as shown by the data and modelling of our paper. For example, Richards (2011) states that H₂O contents of >4 wt.% for intermediate to silicic magmas are required. More recently, Rezeau & Jagoutz (2020) show that the initial H₂O content of basaltic arc magmas is what is important (i.e. > 2 wt. H₂O) but not necessarily reaching super-wet conditions. Chiaradia (2020) also showed a similar point that arc basalts with normal H₂O contents are the most porphyry fertile but do not suggest these evolve to being super-wet. On the other hand, Loucks & Fiorentini (2023) suggest based on Zr concentrations of porphyry magmas that these have > 9 wt.% H₂O. We thus prefer not to state a clear link between super-wet magmas and porphyry Cu deposits in the introduction, but we do introduce a potential link on L64 of the revised manuscript.

L226-228: The question is not really whether Cu deposits can be associated with super-wet magmas, but if there is a relationship between water content and size and or metal content.

We agree this is an interesting topic regarding a potential relationship between deposit size/metal content and water content. However, we do not think, as discussed in our comments above that an association of porphyry copper deposits and super-wet magmas is not widely accepted as a lot of the previous evidence is circumstantial and our study aims to provide a more robust link between the two processes. We do not observe, based on current data (see Figure 7) a that larger deposits have lower zircon Ti contents than smaller systems.

We have added a couple of sentences to the revised manuscript (L253-255):

Our comparison indicates that low zircon Ti contents are present in porphyry Cu deposits regardless of their size, grade or deposit type. Thus, super-wet arc magmatism exerts a first-order control on porphyry Cu mineralisation.

L234-252: The text requires significant rewording as it is difficult to follow.

This paragraph has been rewritten and the previous contents are now split so that the parts referring the porphyry Cu deposits are in one section and the more general parts about arc magmas are in the subsequent section

L273: This is the first time you refer to the largest systems.

See our comments above.

L322-324: Interesting but very speculative.

True, we have removed this final paragraph because, as the reviewer suggests above, it is the final paragraph of the paper and deters from the key messaging of our work and is highly speculative.

References:

Chiaradia, M., 2020. How much water in basaltic melts parental to porphyry copper deposits?. *Frontiers in Earth Science*, 8, p.138.

Keller, C.B., Schoene, B., Barboni, M., Samperton, K.M. and Husson, J.M., 2015. Volcanic–plutonic parity and the differentiation of the continental crust. *Nature*, 523(7560), pp.301-307.

Lee, C.T.A. and Bachmann, O., 2014. How important is the role of crystal fractionation in making intermediate magmas? Insights from Zr and P systematics. *Earth and Planetary Science Letters*, 393, pp.266-274.

Loucks, R.R. and Fiorentini, M.L., 2023. Early zircon saturation in adakitic magmatic differentiation series and low Zr content of porphyry copper magmas. *Mineralium Deposita*, 58(8), pp.1381-1393.

Richards, J.P., 2011. High Sr/Y arc magmas and porphyry Cu±Mo±Au deposits: Just add water. *Economic Geology*, 106(7), pp.1075-1081.

Rezeau, H. and Jagoutz, O., 2020. The importance of H₂O in arc magmas for the formation of porphyry Cu deposits. *Ore Geology Reviews*, 126, p.103744.

Reviewer #2 (Remarks to the Author):

Review letter of “A zircon case for super-wet arc magmas” by Nathwani and coauthors.

High magmatic water contents are prerequisite for the formation of economically important magmatic-hydrothermal systems, such as porphyry Cu deposits. However, the direct evidence for such super-wet magmas is rarely preserved due to degassing at shallow crustal levels. The authors innovatively used thermodynamic modelling on zircon, a mineral that survives various lower-to-upper crustal geological processes, to quantify the role of water on zircon saturation. Through comprehensive modelling, they demonstrated that increasing water content in magmas leads to low and narrow ranges of zircon crystallization temperatures. The authors use zircon Ti concentrations as a proxy for magma water content and provide compelling evidence for the prevalence of super-wet magmas in porphyry Cu systems. The observed low and homogeneous Ti concentrations in zircons from porphyry Cu deposits suggest a link between super-wet magmatism and mineralization processes, with important implications for ore deposit formation.

The manuscript is well-written with conclusions supported by robust modelling and provides a significant contribution to our understanding of arc magmatism and associated mineralization processes.

Overall, I believe this manuscript represents a strong candidate for publication in Nature Communications. However, I recommend minor revisions to clarify certain points and strengthen the manuscript's overall coherence. Detailed comments and suggestions for improvement are provided below.

We thank the reviewer for their kind words on our manuscript, and for the minor revisions and pointing out small errors we have made. Below, we provide a response to each comment and how we have amended in the revised manuscript.

1. Abstract: it is worth mentioning how to use zircon to pinpoint super-wet magmas, such as low and homogeneous zircon Ti (< 12 ppm, 75th percentile) concentrations, and how this new understanding can help exploration of porphyry Cu deposits to meet the increasing demands of critical metals as the world transitions to green energy economy.

We agree to some in the community this could be useful. We have added in parenthesis a reference to what we mean by low, homogeneous Ti contents as the reviewer suggests as otherwise it is not very clear to the reader. In the abstract we have added:

Integrating our model with the titanium-in-zircon thermometer indicates that zircon crystallisation under deep (~400 MPa), super-wet conditions (>6 wt.% H₂O) is present in many magmatic systems, particularly those forming porphyry copper deposits (75th percentile of titanium contents <10 ppm).

Regarding the second point of the reviewer, we prefer not to add this point regarding critical metals and the green energy economy as it is not something we touch upon elsewhere in the manuscript. Hence, we do not want to confuse the message of the paper as it is written currently in terms of volatiles and magmatism on Earth. As reviewer

1 suggests we try to touch on too many speculative implications and we prefer not to add this point, though we understand it is a topic of importance.

2. The definition of super-wet magmas is inconsistent throughout the MS. In the abstract, it refers to >6 wt% H₂O, but in the main text it is >8 wt% H₂O (e.g. Line 46). Please ensure a consistent definition is used throughout.

This is a good point that has been mentioned also by reviewer 1 and we have corrected two locations in the text to state that super-wet refers to >6 wt.% and not >8 wt.%. One is in the abstract and the other is the L46 that the reviewer here points out.

3. Line 124, change “lower” to “higher” temperatures...

Yes this is a mistake. Corrected.

4. Line 126, delete “observed with differentiation” as it is duplicate.

Agreed. Deleted.

5. Line 130, it is not clear how Fig. 3A shows “systematic reduction in liquidus temperature from 1120 C to 1000 C”. Perhaps rephrase it or label the liquidus temperature on Fig. 3A.

This is a fair point. We prefer here to rephrase rather than labelling the liquidus on Fig. 3A which may over clutter the figure. We have thus removed the word “systematic” as this is not clear from the figure and have specified the exact H₂O contents that the liquidus temperatures correspond to so it is clear for the reader. Also corrected a mistake here as at 8 wt.% H₂O the liquidus should be 1050C not 1000C. See L131-135 of the revised manuscript:

First, increasing initial melt water content of an andesitic magma from 2 to 8 wt.% H₂O (at elevated pressure to ensure all H₂O is dissolved) leads to a reduction in liquidus temperature from 1120°C at 2 wt.% H₂O to 1050°C at 8 wt.% H₂O and a corresponding displacement of temperature-melt fraction paths to lower temperatures (Fig. 3A).

6. Line 135-136, the statement of “with a shift of approximately +30 C in zircon saturation temperatures from 2 to 8 wt% H₂O_i (Fig. 3B)” is not consistent with the figure. Fig. 3B shows that zircon saturates at 850°C for 2 wt.% H₂O_i and 790°C at 8 wt.% H₂O_i when the initial Zr content of the magma is fixed at 100 ppm. So it is a shift of -60 C (not + 30 C) from 2 to 8 wt% H₂O_i.

Yes this is a mistake and has been corrected to -60C in the revised manuscript.

7. Line 138, unclear what “high H₂O will do so at lower temperatures...” mean. Please rephrase.

We agree this sentence isn't completely clear and have rephrased as follows (see L141-2 of the revised manuscript):

One key finding of our model is that magmas with higher H₂O will crystallise zircon at lower temperatures (Fig. 3)

8. Line 181, please change “<780 C” to “<790 C” to be consistent with Line 122.

Corrected.

9. Line 193, “the slope of the crystallinity-temperature curve steepens; Fig. 3A”. It appears Fig. 5A rather than 3A should be cited here.

We are not sure here about the reviewer’s comment because the slope of the crystallinity-temperature curve is on Figure 3A rather than Figure 5A.

10. Line 194-196, “Increasing H₂O_i causes zircon to appear as an earlier phase in the crystallisation sequence relative to other phases in the thermodynamic model (i.e., at lower melt fraction; Fig. 2B and D”. The citation of Fig. 2B and D is apparently incorrect. Are you referring to Fig. 5B? If so, lower melt fraction should be higher melt fraction if zircon crystallized earlier than other phases.

Thanks for pointing this out – this is indeed referring to Fig. 5B and should be “higher melt fraction”. These have been corrected in the revised manuscript (L204-206), as below:

*Increasing H₂O_i causes zircon to appear as an earlier phase in the crystallisation sequence relative to other phases in the thermodynamic model (i.e., at **higher melt fraction; Fig. 5A and B**),*

11. Line 522, NIST-612 data are missing in Supplementary Table 9.

This was a mistake as it was actually only 91500 that was measured as a secondary standard. We have deleted the references to NIST-612. We note that our secondary standard trace element data (91500) is in exceptional agreement with reported data from Wiedenbeck et al. (2004) (see Supplementary Table 9), hence we are confident our zircon trace element data is robust.

12. Line 523, delete “and GJ-1” after Rak

Corrected.

13. Supplementary Table 1, the ablation pit depth and spot diameter should be in μm rather than mm.

Both corrected in the revised table.

14. Supplementary Table 2, please add the ²³⁸U/²⁰⁶Pb ratios and uncertainties so that Tera-Wasserburg Concordia can be plotted.

Supplementary Table 2 contains the ²⁰⁶Pb/²³⁸U ratio which can simply be converted to ²³⁸U/²⁰⁶Pb by taking the reciprocal. We therefore do not think it adds any further information to add this data and, moreover, the Tera-Wasserburg plot can be generated with the data that is already presented (NB that isoplotR can take ²⁰⁶Pb/²³⁸U as an input to generate a Tera-Wasserburg).

Cheers,

Reviewer #3 (Remarks to the Author):

Your ms is an excellent contribution to very important, timely issues. I believe that it can be made even better with some rather simple, minor tuning, as I'll describe below and in comments and a very few edits in the text.

Strengths are evident: you simultaneously address Big Geoscience Questions, both applied: generation of critical metal (Cu + others) deposits – and fundamental: arc magmatism, role of H₂O, construction of ~permanent crust. And you do so carefully and convincingly, with a novel, ambitious but well-thought-out combination of modelling techniques and regional and global data sets. I expected to be skeptical, at least in part, but I find the discussion of methods and interpretations to be quite satisfying. And the paper is very well-written and illustrated and has good supporting documentation.

We are pleased by the positive comments by the reviewer and for the numerous suggestions for minor improvements (particularly clarifications in our modelling). We have responded to each comment by the reviewer below and specified how we have or have not modified the manuscript in response.

As I said above, I do have some suggestions, mostly regarding clarity and/or expanding on a couple of things. I'll start with a couple of very minor suggestions about supplementary material (which is for the most part exemplary):

◇ Fig S2: Are bold symbols sample means and muted symbols individual measurements? I suspect so, but make this clear in caption.

Yes, this needs to be made explicit. We have added the following sentence into the figure caption for Figure S2:

Translucent symbols are zircon Ti concentrations in individual crystals and opaque symbols indicate the mean Ti concentration for each sample.

◇Supplementary Tables: give them abbreviated names in the tabs, to make it easier for readers to find data of interest.

Good suggestion – we have added abbreviated names in the Excel tabs of the revised supplementary material:

Contents	1. LA-ICP-MS Metadata	2. Zircon TEs	3. WR Data	4. Zircon Ti Comp	5. Porphyry zircon Ti	6. Volcanic WR Zr	7. Porphyry WR Zr	8. U-Pb standards	9. TE standards
----------	-----------------------	---------------	------------	-------------------	-----------------------	-------------------	-------------------	-------------------	-----------------

◇Sup. Table 6 (“Whole-rock Zr compilation for different melt H₂O contents’) is, I believe, referred to as Sup. Table 7 (“Compilation of whole-rock compositions from porphyry Cu districts”) in Figure 2 caption.

Corrected.

I have inserted a number of comments – some of which might lead to minor modifications, others are thoughts that come to my mind while reading your text. Also a very few suggested small edits (wording, punctuation).

Below I expand on four comments, keyed to the text:

Line 206, Note #1: I would recommend expanding and clarifying a bit -- what is “Ti-free?” All minerals have some Ti, though negligible in qtz & feldspar (and zircon). But mafic silicates – especially hydrous mafic silicates- have a fair amount, many accessories (oxides, titanite, chevkinite) have a lot. BUT R-MELTS doesn’t model these minerals well, if at all. How do you assess the role of significantly Ti-bearing minerals (“non-Ti-free”)? You cover this pretty well in your Methods section, but when you address the Ti approach in the main text, I think it would be to expand a bit on how you’ve dealt with it or at least sent reader to Methods.

The use of “Ti-free” here was a bit misleading and instead the correct terminology would be “Ti-poor” since, they are not completely Ti-free as the reviewer suggests. This has been corrected in the revised sentence (L216). We also add reference to “Ti-rich” minerals and give examples for both, so that it is absolutely clear what we mean as the interpretation could be ambiguous otherwise:

*We find that increasing H_2O_i causes lower melt $aTiO_2$ at zircon saturation due to the suppressed crystallisation of **Ti-poor phases (e.g. plagioclase)** and promoted crystallisation of **Ti-rich minerals (e.g. magnetite and amphibole)** at higher temperature, from $aTiO_2 = 1.0$ at 2 wt.% H_2O to $aTiO_2 = 0.4$ at 8 wt.% H_2O .*

Indeed, also rhyolite-MELTS models these Ti-bearing minerals quite poorly and hence the implementation for the Borisov and Aranovich model for rutile solubility. We have already referred the reader to the Methods section here (see L215 of revised manuscript) where we describe in detail this approach which might be too much detail for general readership.

Line 528, Note #2: It seems like a stretch to say that modelling was done simply using Rhyolite-MELTS. What you describe in this section is not modelling of zircon saturation – you don’t mention zircon at all except in this sentence. Results of what you describe here can (and are) used as input into zircon saturation modelling, and you present this approach in the following sections.

We agree with the reviewer it was a bit misleading in the way things are phrased as it gives the impression that the zircon saturation modelling was solely achieved using rhyolite-MELTS which is not the case. This sentence (originally on L528) has thus been rewritten (see L553-554 of revised manuscript):

Temperature-crystallinity curves for different initial water contents in intermediate-felsic arc magmas was modelled using rhyolite-MELTS^{32,33}.

We also adjusted the title of this subsection (L552) from “Rhyolite-MELTS modelling of zircon saturation” to “Modelling of zircon saturation as a function of initial melt H_2O ” as this also could have caused confusion.

Line 649 (Fig. 2), Note #3: Your SiO_2 -Zr data are not for magmas, at least not strictly – they are whole rock compositions, which, especially for the intrusive rocks, could differ significantly from magma compositions. And the H_2O estimates in Sup Table 6 (not 7) at least mostly appear to be based on melt (not magma). I think that this is a very informative figure, but I would augment the caption to make clearer what the data (SiO_2 -Zr, H_2O) are based on.

This is a good point from the reviewer as the fact that these are whole-rock compositions in Figure 2 is not specified at all, and the methodology behind which the H₂O estimates are made is also not provided (without requiring the reader to look at the Supplementary material). We have thus now specified they are whole-rock compositions and that the H₂O estimates are based on melt inclusion/experimental constraints. The updated figure caption (now found on L676) is as follows:

Figure 2. Zirconium systematics for magmas with a range of H₂O contents. Whole-rock Zr systematics for mid-ocean ridge lavas²⁹, arc magmas and porphyry Cu deposit-related igneous suites. For arc magmas, the compositions are separated by lower (2-4 wt.%) and higher (4-8 wt.%) H₂O where dissolved H₂O is estimated based on the maximum of available melt inclusion data or from experimental constraints (see Supplementary Table 6). Points show the binned mean of Zr contents per 2 wt.% SiO₂ and 2 s.e. uncertainties. No data could be found for rhyolites (>70 wt.% SiO₂) for arc magmas with 2-4 wt.% H₂O.

Line 689 (Fig. 5), Note #4: I think that this figure needs to be clarified (caption and/or y-axis title) I think that Y values are confusing: what does “Fraction of total zircon crystallized” mean? Taken literally, to me at least, that value should increase monotonically down-T (no valleys in the trend) and reach 1.0 at the solidus. Is the Y value actually the fraction of zircon crystallized within a finite T range? Or what?

We agree here that some clarification is required to make this clear to the reader. What the reviewer describes here would be the *cumulative* fraction of total zircon crystallised whereas what we refer to is the fraction of total zircon crystallised which in total sums to 1 (i.e. at a given temperature it is the fraction of total zircon that crystallised at that temperature relative to the total zircon crystallised at all temperatures).

We have added an additional sentence to the figure caption (see L720-722 of the revised ms):

The fraction of total zircon crystallised refers to the mass of zircon crystallised at each model increment divided by the total mass of zircon crystallised from the magma.

All in all, an excellent, thought-provoking paper! I enjoyed reading it and look forward to seeing it published!

--reviewed

Line by line annotations made on the PDF of the manuscript:

L35: If I'm reading Fig 1 correctly, it seems to suggest that primitive basalts have ~0-5% H₂O, but dominantly 0-3 (not 2-4). Am I misreading it?

Sorry for the confusion here – the 2-4% is referring to what people typically regard to be dissolved in a primitive arc magma, and does not refer to the melt inclusion data shown on Figure 1. We have rewritten this sentence to make this clearer, and referenced the paper of Plank et al. 2013 and used 4% as the accepted primitive H₂O composition (see L35-39 of the revised ms):

This restricted range of water contents in arc melt inclusions, regardless of the extent of chemical differentiation, is at odds with the water contents produced by fractional crystallisation of a typical primitive basalt composition (4 wt.% H₂O) which yields much higher H₂O contents of 6-11 wt.% H₂O¹⁻³ (Fig. 1).

L45: Here, or elsewhere in your Intro, it seems that a citation of Ian Carmichael's andesite aqueduct (2002) would be appropriate.

Yes agreed, we have added that to this sentence, when we refer to the suppression of plagioclase crystallisation in some magmas as evidence for super-wet conditions. Carmichael (2002) suggest the absence of plagioclase in some lavas from Colima requires > 6 wt.% H₂O. See reference number 18.

L68: prior to zircon saturation.

Good point, we have added "prior to zircon saturation" into the parentheses

L69-71: The retained evolved melt will inevitably saturate in zircon saturation even in deep intrusions as it approaches the solidus; how much zircon is retained at depth will depend on how much of this zircon-saturated melt is retained.

Yes this is very true. In this sentence we are stating that it is observed in "significant" quantities in deep intrusions too and may provide a source for zircon in the upper crust. This is often not considered as many assume all zircon in upper crustal rocks crystallises in situ, thus we wanted to review this in the introduction.

L124: ?? higher ??

We have corrected this (also brought to our attention by reviewer 3) in the revised ms

L167: Five have 200-350, but one of the six has ~130

True - thanks, we have corrected this to 130-350 ppm

L169-171: I think it would be useful to show this relationship (SiO₂ vs Zr) for the Central Chile rocks (how the relationship varies with time)

We agree this would be a useful figure, which lends further support to what is shown in Figures 2 and 3. We have added this as a supplementary figure (Supplementary Figure 3). We do not include in the main text as a similar point is made in Figs. 2 and 3.

L195-196: I don't think there are Figs 2B and D. In fact, I don't there's a "D" in any figure (text or supplement)

This was a mistake retained from a previous draft (also noticed by reviewer 2), which should be Figure 5A and 5B. This has been corrected.

L206: I would recommend expanding and clarifying a bit (see Note #1)

We have responded to this comment above in response to Note #1

L209: insert ")"

Inserted

L240: You might note the low Ti in Sierra Nevada, and maybe Japan, as well.

Agreed good point. We have specified this now in L276-279:

When compared broadly with other magmatic arcs (Fig. 7), the ranges in Ti values are similar to those observed in the Sierra Nevada (USA) and Japan, whilst the Coastal Cordillera (Chile) show higher and more variable zircon Ti (Fig. 7; ref⁵²).

L263: I'm not sure that you need this qualifier: I believe that the statement remains true if you remove "typically."

Agreed – we have removed the “typically”

L264: Same comment: isn't this more than a tendency?

Also agree – we have removed the “tend” to

L307: This refers to the 15 km depth, right? (As written, what "Such..." refers to isn't clear.)

We have deleted the "such" and specified mid-crustal in a previous sentence to make it clear what we are referring to, see L317 of the revised manuscript.

L512: minerals

Corrected

L528: Seems like a stretch to say that modelling was done simply using Rhyolite-MELTS (see note #2).

We have responded to this in our reply to Note #2

I understand the general logic of this choice, but the specific rock name you use seems very odd. There are plenty of rocks of andesitic COMPOSITION in Adamello, but they are not andesite (aphanitic; generally taken to be volcanic, though could be applied to hypabyssal rocks of this composition.)

Yes this is true the name we give to this is a bit misleading, we meant that the rock is *intermediate* in composition when stating "andesitic" but not necessarily fine grained. We also emphasise that applying to hypabyssal rocks as the reviewer states is better for our purposes of modelling porphyry dykes

We have thus replaced the word "andesitic" with "intermediate" in the revised manuscript – see L556.

L559: Basically, Rhyolite-MELTS can't be used for hydrous minerals (notably amphiboles and micas). If it does show up, it's probably best to ignore it (or, better, to suppress it). And then to acknowledge that this will have an effect on modelling Zr concentration, but that effect is minor (as you already do).

Yes, indeed we agree. We very rarely see it appear as a phase and thus it has a very minor effect on Zr concentrations of the modelled melt. As the reviewer states, we already acknowledged this so we have not changed anything further in the manuscript.

L561: State what "data" - this is unclear. (Inferred mineral Kds as $f(T)$, right?)

Yes we refer to mineral-melt partition coefficients for Zr (Kds) as a function of T. We have now specified this (L586 of the revised manuscript):

Data for D_{Zr} as a function of temperature were compiled for amphibole, clinopyroxene, plagioclase, biotite and ilmenite (Supplementary Fig. S1).

L573-574: Interestingly (to me at least), it turns out that Watson & Harrison (1983) yields essentially identical results to Crisp & Berry for melt water contents above ~4 wt%. This actually lends credence to C&B (as well as to W&H for typical silicic, relatively H₂O-rich, magmas)(note that in Fig 2 caption you mention that "No data could be found for rhyolites (>70 wt.% SiO₂)" with <4 wt% H₂O). (C&B exaggerate the very slight difference in their paper.)

For your purposes, where you are comparing "wet" to "dry" magmas (<4% H₂O), however, there's definitely an advantage to using C&B.

We agree with the reviewer that there is an advantage to using C&B which is calibrated on a broader range of melt compositions than the original zircon solubility model of W&H. There appears to be nothing further for us to change based on this comment.

L588: Or with natural materials (notably glasses)

True. We have also added “natural glasses” to this sentence.

L590: I'm quite sure that MELTS diversion from "truth" is much more connected with melt fraction (and thus with H₂O) than with T. Is this not what you find?

We do agree with the reviewer on this, after comparing the liquid line of descent versus temperature and melt fraction it appears the departure from normality occurs at a more constant melt fraction rather than temperature.

This may at first suggest that a parameterisation of optical basicity as a function of melt fraction may be more robust, as opposed to temperature as we do in the manuscript. However, considering the plot above a calibration of optical basicity vs melt % (<60%) would produce very low optical basicity values which are far more negative than encountered in natural rocks. Whereas our parameterisation of optical basicity vs temperature produces values within an expected range.

L624: As I guess you realize, this is unrealistic because it's impossible for TiO₂, and a(TiO₂), to fall to zero. Your treatment below is probably as good an approach to avoid this problem as any.

Yes we agree this is unrealistic and hence we preferred the Borisov and Aranovic rutile solubility model. We have rephrased this sentence to make it clear that it is unrealistic:

We also find that at low temperatures (<800°C) the values of aTiO₂ **can unrealistically collapse to zero** as TiO₂ reaches **very low concentrations** in the melt in the melt.

L649: Note #3 (magma vs whole rock? H₂O determination?)

We respond to this in our reply to Note 3.

L651: I think that you're referring to Supplementary Table 6, right?

Yes this was also noted by reviewer 2 and we have corrected to Supplementary Table 6

L662: These SiO₂ %s are calculated based on normalized major oxides ("volatile-free"), right?

Yes correct, we have now specified that is calculated on an anhydrous basis. See L692 of the revised ms.

L689: I think that this figure needs to be clarified (caption and/or y-axis title) (See Note #4)

We have clarified this figure caption, see our response to Note #4 for more information

L715: This table is restricted to porphyry systems. You probably should also list Table S4 for the other data summarized here.

We have now stated "Supplementary Tables S4 and S5" in the revised figure caption.

REVIEWERS' COMMENTS

Reviewer #1 (Remarks to the Author):

Dear authors and editor,

Thank you for giving the opportunity to go through the revised manuscript. I am pleased to say that I think that it is greatly improved, especially as it focusses more on the real advances in the field and has moved away from some of the more speculative topics, which are interesting but maybe a bit distracting.

I have no objections to the paper being published. However, I have provided some additional notes of an annotated draft that I have uploaded on the server, where I suggest a shift in focus, at least in the abstract and in the introduction. In fact, I think that at the moment the abstract does not do justice to the work. It just focusses on water contents (2, 4, 6, > 6 wt%...?), and Zr values, rather than capturing what in my opinion is the most elegant and interesting take home message from the work, ie the relationship between water content and Ti values.

I have indicated in the annotated draft numerous topics that should make their way on to the abstract, and suggestions for a change in focus to enhance the impact of the study, which otherwise has the risk not to be cited in the future for the right reasons...

I congratulate the authors for an excellent piece of work. Please feel free to accept or disregard my suggested edits and comments as you see fit. I personally think that a focus on titanium would greatly enhance impact.

Reviewer #3 (Remarks to the Author):

As I said of your initial submission: "Your [initially submitted] ms is [was already] an excellent contribution to very important, timely issues. I believe[d] that it can[could] be made even better with some rather simple, minor tuning, as I'll describe below and in comments and a very few edits in the text." You have tuned it admirably. Your responses to all of my comments are excellent and thorough. I have no further suggestions: in my opinion your ms is now in fine shape for publication.

Reviewer #1 (Remarks to the Author):

Dear authors and editor,

Thank you for giving the opportunity to go through the revised manuscript. I am pleased to say that I think that it is greatly improved, especially as it focusses more on the real advances in the field and has moved away from some of the more speculative topics, which are interesting but maybe a bit distracting.

I have no objections to the paper being published. However, I have provided some additional notes of an annotated draft that I have uploaded on the server, where I suggest a shift in focus, at least in the abstract and in the introduction. In fact, I think that at the moment the abstract does not do justice to the work. It just focusses on water contents (2, 4, 6, > 6 wt%...?), and Zr values, rather than capturing what in my opinion is the most elegant and interesting take home message from the work, ie the relationship between water content and Ti values.

I have indicated in the annotated draft numerous topics that should make their way on to the abstract, and suggestions for a change in focus to enhance the impact of the study, which otherwise has the risk not to be cited in the future for the right reasons...

I congratulate the authors for an excellent piece of work. Please feel free to accept or disregard my suggested edits and comments as you see fit. I personally think that a focus on titanium would greatly enhance impact.

Best wishes

We are pleased the reviewer thinks our revised manuscript is now better focused and more impactful, thanks to their previous comments. We also thank the reviewer for the new suggestions on the abstract. We agree with the comment that it would be good to place focus on the relationship between high water contents and Ti concentrations of zircon as this is a big novel takeaway message that could be used by future researchers. The reviewer also noted on the annotated draft other aspects of our study should be emphasized in the abstract: using zircon Ti to track melt water contents in the study, and the relationship between low zircon Ti and the formation of the world's largest copper deposit in Central Chile.

We have re-written the abstract to take these points into account, whilst still keeping it broad for the multi-disciplinary audience of the journal. The abstract has been circulated amongst co-authors and shown to colleagues unrelated to the study to ensure it is optimal.

L12-13: Repetition

This sentence is no longer present in the revised abstract

L49: Logic gap here. Connect these two paragraphs more smoothly

We have now added a smoother transition between these two paragraphs by adding: "Alternative methods provide..." in reference to the melt inclusion approaches described in the previous paragraph

L57: Consistency...inverted commas?

We have added inverted commas here

L59-60: Is the predicament that it is that when and where this happens that mineralisation occurs?

This is indeed possible though difficult to provide evidence for. No changes required here.

L103-104: This should be emphasised also in the abstract

Yes, we have now specified that aspect of fingerprinting super-wet magmas using zircon Ti concentrations in the abstract, though we do not mention low bulk-rock Zr because this is less novel as it has been shown by the reviewer in a recent paper and due to the 150 word limit for the abstract.

L144-147: This should be the take home from the paper

Yes we agree that the application of zircon Ti to show water contents should be a key take home message which is now made clear in the abstract. Though we prefer not to refer to the "where a complete differentiation series of cogenetic magmas is not available" as this is too specific for the abstract of a broad journal.

L182: How about this as a title?

Though we agree for a more discipline specific journal this would be an excellent title, for this broader impact journal we think the current title is more suitable.

L227-232: This is great...but again, how about making this the focus in the abstract?

Yes we have now added a clear sentence in the abstract that low, homogeneous zircon Ti concentrations are linked to high melt H₂O:

We demonstrate that super-wet magmas crystallise zircon with low, homogeneous Ti concentrations (75th percentile <10 ppm) due to a decrease in zircon saturation temperatures with increasing melt H₂O.

L327-334: Also this should be included in the abstract

Yes we have now included all the details into the revised abstract. Particularly now with a sentence specifying the link between low zircon Ti, super-wet magmas and the formation of the world's largest porphyry copper deposit:

We find that zircon Ti concentrations record a transition to super-wet magmatism in Central Chile immediately before the formation of the world's largest porphyry copper deposit cluster at Rio Blanco-Los Bronces

Reviewer #3 (Remarks to the Author):

As I said of your initial submission: "Your [initially submitted] ms is [was already] an excellent contribution to very important, timely issues. I believe[d] that it can[could] be made even better with some rather simple, minor tuning, as I'll describe below and in comments and a very few edits in the text." You have tuned it admirably. Your responses to all of my comments are excellent and thorough. I have no further suggestions: in my opinion your ms is now in fine shape for publication.

We thank the reviewer again for their constructive feedback on the manuscript and are pleased they believe it is ready for publication